# Planar cell polarity signalling coordinates heart tube remodelling through tissue-scale polarisation of actomyosin activity

Anne Margarete Merks [1], Marie Swinarski [1], Alexander Matthias Meyer [1], Nicola Victoria Müller [1,2], Ismail Özcan [1], Stefan Donat [3,4], Alexa Burger[5], Stephen Gilbert[6], Christian Mosimann [5], Salim Abdelilah-Seyfried [3,4] & Daniela Panáková [1,7]

Development of a multiple-chambered heart from the linear heart tube is inherently linked to cardiac looping. Although many molecular factors regulating the process of cardiac chamber ballooning have been identified, the cellular mechanisms underlying the chamber formation remain unclear. Here, we demonstrate that cardiac chambers remodel by cell neighbour exchange of cardiomyocytes guided by the planar cell polarity (PCP) pathway triggered by two non-canonical Wnt ligands, Wnt5b and Wnt11. We find that PCP signalling coordinates the localisation of actomyosin activity, and thus the efficiency of cell neighbour exchange. On a tissue-scale, PCP signalling planar-polarises tissue tension by restricting the actomyosin contractility to the apical membranes of outflow tract cells. The tissue-scale polarisation of actomyosin contractility is required for cardiac looping that occurs concurrently with chamber ballooning. Taken together, our data reveal that instructive PCP signals couple cardiac chamber expansion with cardiac looping through the organ-scale polarisation of actomyosin-based tissue tension.

[1] Electrochemical Signaling in Development and Disease, Max Delbrück Center for Molecular Medicine in the Helmholtz Association, Berlin-Buch 13125, Germany. [2] Department of Clinical Pharmacology and Toxicology, Charité—Universitätsmedizin Berlin, Berlin 10117, Germany. [3] Institute for Biochemistry and Biology, Animal Physiology, University Potsdam, Potsdam 14476, Germany. [4] Institute for Molecular Biology, Hannover Medical School, Hannover 30625, Germany. [5] Institute of Molecular Life Sciences, University of Zürich, Zürich 8057, Switzerland. [6] Mathematical Cell Physiology, Max Delbrück Centre for Molecular Medicine in the Helmholtz Association, Berlin-Buch 13125, Germany. [7] DZHK (German Centre for Cardiovascular Research), Partner Site Berlin, Berlin 13125, Germany. These authors contributed equally: Anne Margarete Merks, Marie Swinarski. Correspondence and requests for materials should be addressed to D.P. (email: daniela.panakova@mdc-berlin.de)

Most of the organ systems of the animal body arise from simple epithelial tubes. While organs such as the lung or the pancreas undergo branching morphogenesis, others including the brain and the heart remodel into more complex forms. The linear heart tube (LHT) emerges during vertebrate development as a transient structure composed of an inner endothelial tube surrounded by a single-cell epithelial layer of cardiac muscle. The LHT forms in humans at 20–22 days, in mouse at 8 days, and in chick at 1.5 days of embryonic development, while in zebrafish the LHT forms already at 22 h of post fertilisation (hpf)[1–3]. The LHT early on displaces leftward relative to the dorsal midline of the embryo, followed by bending and twisting during cardiac looping[1–5]. During this process, cardiac chambers start forming through the process of cardiac chamber ballooning that results in the distinct asymmetries between the atrial and ventricular chambers[2, 6].

The current two-step model of chamber remodelling is based on the anatomical and quantitative reconstruction of cell size and proliferation[7]. In the two-step model, the LHT is formed by slowly proliferating cardiomyocytes with their cell size gradually increasing on the ventral side of the tube[7]. This regional increase in cell size[7,8] and the subsequent differential hypertrophic growth has been demonstrated experimentally and by computational modelling to be the driving force behind cardiac looping and chamber ballooning[7,9]. The consequence of these complex morphogenetic processes is the emergence of the atrio-ventricular junction (AVJ), and the formation of the atrium and the ventricle that in zebrafish acquire characteristic bean-like shape morphology with inner (IC) and convex outer curvatures (OC). Importantly, the initial chamber ballooning and looping occurs without any cell proliferation, and the chambers expand by accrual of myocardial cells from the second heart field (SHF), shaping the sinus node at the venous pole and the outflow tract (OFT) at the arterial pole[2,3,10].

Considerable efforts have been dedicated to determine genetic programmes that contribute to cardiac chamber specification and morphogenesis[2,3,10]. Many signalling events and transcription factor networks regulating cardiac progenitor determination, lineage commitment, or chamber-specific myocyte differentiation have been identified through genetic screens and loss-of-function analysis in mouse, chick, zebrafish and in vitro differentiation assays[3,10]. Detailed retrospective clonal analysis in the mouse has revealed that the expansion of cardiac chambers is coordinated through oriented clonal growth consistent with the left ventricle bulging from the outer curvature of the LHT[11]. Nonetheless, both the underlying signalling and the cellular mechanisms that drive the chamber formation and the LHT remodelling remain unclear.

Planar cell polarity (PCP) pathway, a non-canonical branch of Wnt signalling, refers to the mechanisms providing directional information at the local as well as at the global scale; at the local level, cells orient themselves with respect to their neighbours, at the global level cells align in a cooperative manner with a specific orientation within a larger field of cells[12–17]. The core PCP pathway components comprise the transmembrane proteins Frizzled (Fzd) and Vang-like (Vangl) and their cytoplasmic binding partners Dishevelled (Dvl) and Prickle (Pk)[12–17]. While Fzd and Dvl are described as positive regulators of PCP signalling, Pk and Vangl function antagonise the signalling system intra- as well as intercellularly[12–14]. PCP signalling is indispensable for several morphogenetic processes during organ development, for instance in neural tube closure as well as in lung or kidney branching[18–20]. Although Wnt non-canonical ligands and all core PCP components are expressed in the heart[21–25], and mutations in several pathway components lead to congenital heart disease associated with defects in outflow tract remodelling[26], the precise

role of PCP during cardiogenesis remains incompletely understood.

Here, we show that cardiac chambers expand through epithelial remodelling driven by cell neighbour exchange. We found that the non-canonical Wnt/PCP pathway, guides the morphogenesis of the early myocardium by restricting local actomyosin contractility. We discovered that PCP coordinates localised actomyosin activity at two distinct levels: first, PCP affects actomyosin activity locally at the cellular level and may alter the efficiency of cell neighbour exchange; second, PCP planar-polarises actomyosin at the tissue-scale by limiting its activity to the apical membranes of the distal ventricle and the outflow tract cardiomyocytes. We propose that such polarity in tissue tension generates the mechanical forces required for bending of the LHT during cardiac looping and assists in bulging of the cardiac chambers. Indeed, we found that loss of PCP signalling leads to complete inability of the LHT to undergo cardiac looping in vitro, and causes impaired looping in vivo. Our findings provide insights into the cellular mechanisms underlying cardiac chamber formation and looping, and connect these processes to instructive PCP signals.

## Results

**The LHT undergoes epithelial remodelling.** The bean-like cardiac chamber morphology with its distinctive convex curvatures has been at least in part attributed to the regionally restricted changes in cell shapes between OC and IC (Fig. 1a)[8]. In line with previous reports, we measured that OC cells are larger and less cuboidal at 54 hpf compared to IC cardiomyocytes with an average area of 104 $\mu m^2$ and circularity value of 0.53, while the IC cells are smaller and rounder with the average area of 83 $\mu m^2$, and the circularity value of 0.6, respectively (Fig. 1b). The bean-like ventricular morphology is characterised not only by the regional cell shape changes, but also by distinct cell orientation (Fig. 1a). We analysed cell orientation within the ventricular AV, IC, OC and OFT regions to determine the angle on a scale ±90° between the orientation of the cell and the local ventricular surface tangent (Fig. 1a). In wild-type hearts, the cells of the IC and OC are oriented around 0°, whereas those at the AV-junction and within the OFT region are oriented around 90° relative to the ventricle outline (Fig. 1c). For statistical analysis of angle distribution, we performed Rayleigh's test to assess significance of the mean direction by using a concentration parameter of a circular distribution; here, low values of Rayleigh's $R$ represent highly concentrated values with little distribution. We focused on the OFT region as ventricular chamber expansion depends on the accrual of cells from the SHF-derived pool of cardiac progenitors through the arterial pole, and we observed that the mean angle of the OFT cardiomyocyte orientation in wild-type hearts is 92.27° with low distribution (Fig. 1c). Taken together, these data confirm that the presence of regionally specific cardiomyocyte cell shapes and cell orientations within the ventricle accompanies cardiac chamber formation in zebrafish.

Cell rearrangements in epithelial as well as non-epithelial tissues occur through transition states in which four (T1-transition) or more cells (rosette) converge into sharing a single-cell boundary[27,28]. The subsequent resolution and formation of the new cell boundaries provide an efficient mechanism for the rearrangement of cells within a single-layered epithelium[27–29]. As the developing myocardium is initially also a single-layered epithelium[2], we hypothesised that chamber ballooning involves an active tissue remodelling process. To visualise the cell neighbour exchange in vivo, we performed live imaging from 26 to 31 hpf as the LHT initiates looping. We used zebrafish embryos of $Tg(myl7:lck\text{-}EGFP)^{md7130}$, which express membrane-tethered

EGFP under the myocardium-specific *myl7* promoter (Fig. 1d, Supplementary Movie 1), and to suppress motion artefacts we injected *myl7:lck-EGFP* embryos with the established *tnnt2a^{ATG}* morpholino that blocks heart contraction[31]. We detected the formation of transition states within the myocardium (Fig. 1d, arrow in inset) with cells exchanging neighbours and forming

new cell boundaries over time (Fig. 1d, arrowhead in inset). Although the time required for the resolution of the transition state is most likely skewed in our non-beating hearts as active contractility affects myocardial properties[8], our data indicate that cellular rearrangements do occur during cardiac chamber ballooning.

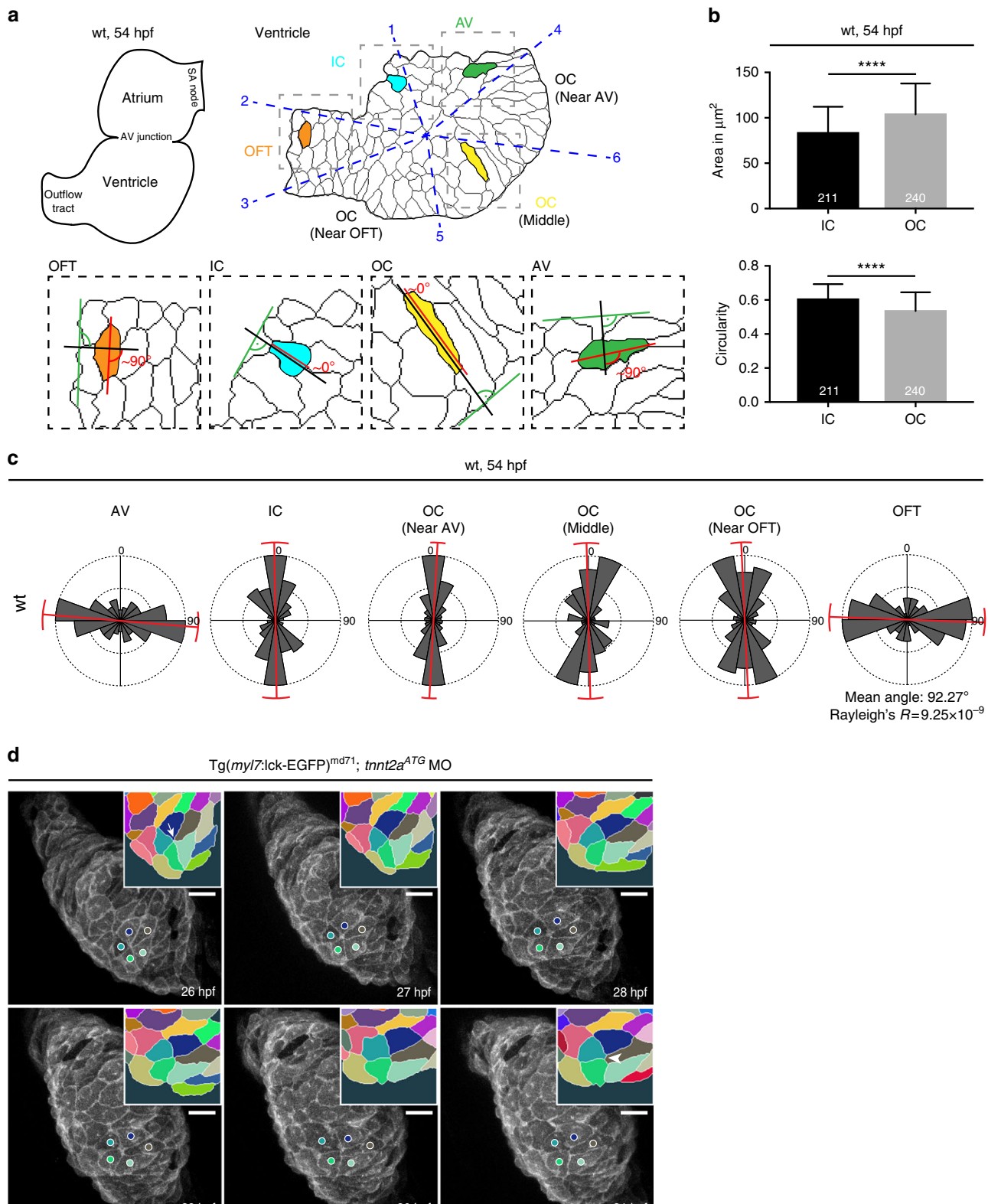

**Fig. 1** Epithelial remodelling of the LHT during chamber formation. **a** Scheme of two-chambered zebrafish heart at 54 hpf (left) next to the smoothened outlines of a ventricle depicting shapes of the cardiomyocytes (right). According to six defined anatomical landmarks (number 1–6 in blue) the ventricle is segmented into six regions (blue dashed line) connected with the ventricular centroid, counter-clockwise: between 1–2: inner curvature (IC), 2–3: outflow tract (OFT), 3–5: outer curvature (OC) (near OFT), 5–6: OC (middle), 6–4: OC (near AV) and 4–1: atrio-ventricular junction (AV). In insets, the scheme depicts the cell orientation in four different segments defined as the angle between the cell elongation axis (red) and a 90° angle (black) placed above a tangent (green) on the smoothened outline of a ventricle. Cell orientation of an OC (yellow) and IC (turquoise) cell is ~0°, cell orientation of an OFT (orange) and an AV cell (green) is ~90°. **b** In wild-type (wt) hearts, OC cardiomyocytes are large and elongated ($n = 240$, area = 107 µm$^2$, circularity = 0.53), while IC cardiomyocytes are small and rounded ($n = 211$, area = 83 µm$^2$, circularity = 0.6). Hearts analysed, $n = 8$. Means ± s.d. ****$P < 0.0001$, unpaired $t$-test with Welch correction. **c** Based on schematic in **a**, AV (cells analysed, $n = 92$) and OFT ($n = 86$) cardiomyocytes assume ~90° angle, IC ($n = 57$) and OC ($n = 295$) cells assume ~0° angle. 8 hearts analysed. Variance of angle distribution is labelled in red. **d** Whole embryo time-lapse imaging of a resolving transition state in a *tnnt2a*-deficient heart expressing membrane-associated EGFP under *myl7* promoter. At 26 hpf, the LHT displays a transition state of five cardiomyocytes sharing a common boundary (arrow in inset). During the following 5 h the transition state resolves with newly forming cell junction (arrowhead in inset). Transition state is colour-coded in the insets, corresponding cells marked with coloured circles. Scale bars, 20 µm

To define the crucial period of cardiac chamber remodelling, we quantified the number of transition states within the ventricle at different stages during chamber formation. At 26 hpf, the LHT myocardium showed a considerable number of transition states (Fig. 2a, d). The number of transition states decreases by half after formation of the ventricle and atrium at 54 hpf (Fig. 2b, d). By 72 hpf, when cardiac chamber formation and heart looping is completed, the dynamics of cellular rearrangements do not significantly change (Fig. 2c, d, ordinary one-way ANOVA). Combined, our data indicate that epithelial remodelling via cell neighbour exchange occurs concomitantly with cardiac chamber formation in the critical period between 26 and 54 hpf.

**PCP signalling guides cell rearrangements during chamber formation.** The PCP axis of non-canonical Wnt signalling guides cell rearrangements during the morphogenesis of numerous organs[13,32]. Wnt5b and Wnt11 belong to key Wnt ligands required in cardiogenesis that have been described in the context of the PCP signalling pathway[33]. To test the potential requirement of PCP signalling in cardiac remodelling, we used both morpholino knockdown technology and genetic mutants. We verified that morpholino-induced phenotypes recapitulate genetic loss of Wnt5b and Wnt11 ligands as well as PCP core components as per current guidelines[34] (Supplementary Fig. 1). Loss of either *wnt5b* or *wnt11* leads to slight increases in transition states in the myocardium at 54 hpf (Fig. 2e, f, l, Supplementary Fig. 2a). Compared to wild-type controls, the number of transition states increases by 2-fold in *wnt5b*$^{ta98}$ [35]; *wnt11*$^{tx226}$ [36] double-mutants (Fig. 2g, l) or in *wnt5b*$^{ex6}$ MO; *wnt11*$^{ATG}$ MO (Fig. 2l, Supplementary Fig. 2b), suggesting that Wnt5b and Wnt11 act redundantly in affecting cell neighbour exchanges.

To elucidate the function of PCP signalling downstream of Wnt5b and Wnt11 in the regulation of cell rearrangements during cardiac remodelling, we quantified the number of transition states within the myocardium in the absence or with decreased levels of all core components of PCP pathway. We tested both morpholino knockdown and the existing loss-of-function mutants *fzd7a*$^{e3}$ [37] and *vangl2*$^{m209}$ [38]. Compared to wild-type hearts at 54 hpf, the number of transition states was significantly increased in *fzd7a* mutant hearts or *fzd7a*$^{5'UTR}$ MO (Fig. 2h, l, ordinary one-way ANOVA; Supplementary Fig. 2c), and in *dvl2*$^{ATG}$ MO hearts (Fig. 2i, l, ordinary one-way ANOVA). In *vangl2*$^{m209}$ mutant hearts the number of transition states is comparable to wild type, while the reduction of *vangl2* slightly decreased the number of transition states (Fig. 2j, l, Supplementary Fig. 2d); in *pk1a* morphant hearts the number of transition states mildly increased (Fig. 2j–l). Our data demonstrate that remodelling of the cardiac chambers through myocardial cell rearrangements is affected by perturbed Wnt-dependent PCP signalling. Importantly, this conclusion is corroborated by similar results observed with both genetic mutants and morphants.

The effect on the number of transition states at 54 hpf in perturbed PCP signalling might occur due to altered dynamics of transition state formation or their resolution at earlier time points. To distinguish between these effects, we also quantified the number of transitions at 26 hpf, assuming that an increase in transition states already at this early stage would hint at an increased frequency of rosette formation. Indeed, in hearts with decreased levels of *fzd7a*, the number of transition states was increased at this stage compared to wild-type hearts (Supplementary Fig. 2a, b, d). In contrast, reduction of *vangl2* led to slight decrease in the number of transitions (Supplementary Fig. 3c, d). These results suggest that PCP signalling may regulate the efficiency of cell neighbour exchange during myocardial remodelling.

We next asked whether altered cell neighbour exchange results in changes in ventricular morphology. We found that deficient PCP signalling leads to significant misalignment of cells: whereas absence of *fzd7a* resulted in severe disruption of cardiomyocyte orientation, especially in regions close to the OFT (Fig. 3a), loss of *vangl2*, *dvl2* and *pk1a* resulted mainly in increased variability of cell orientation within the OFT region (Fig. 3b–d, Rayleigh's test). In hearts with a complete loss of *fzd7a*, the mean angle of OFT cell orientation shifted to 118.42° with a high variance (Fig. 3a, Rayleigh's test). While the mean angle of OFT cardiomyocyte orientation in *vangl2* mutant or *dvl2* and *pk1a* morphant hearts was not markedly altered, the variability of angles was high (Fig. 3b–d). Taken together, our findings indicate that PCP-dependent LHT remodelling shapes the ventricular chambers.

**PCP affects the contractile actomyosin network.** Tissue remodelling is guided by complex mechanisms that involve force generation through modulation of actomyosin contractility together with cadherin localisation and turnover during cell adherens junction complex assembly and disassembly[39]. PCP signalling affects both actomyosin and cadherins in various developmental contexts[32]. To determine the effects of PCP pathway on actomyosin and cadherin on cell neighbour exchange during cardiac remodelling, we focused on the role of the core Fzd/Vangl module, as these transmembrane proteins act antagonistically in several contexts to initiate intracellular processes[12–14]. We first assessed the localisation and abundance of N-Cadherin, the only classical Cadherin expressed in the developing heart[40], in *fzd7a* and *vangl2* genetic mutants or in *fzd7a*- and *vangl2*-deficient myocardium and compared them to wild-type hearts. While at 54 hpf, aPKC localised to the apical tight junctions, N-Cadherin is present uniformly at the basolateral membranes of wild-type hearts as well as of the hearts lacking *fzd7a* and *vangl2* (Fig. 4a–f, Supplementary Fig. 4a–c). Neither N-Cadherin levels nor its cellular distribution was affected, suggesting that the PCP pathway does not affect the N-cadherin steady state during cardiac chamber remodelling.

Specific changes in cell shape can act as a proxy to determine the local changes in the actomyosin contractile network. Visualisation of sparsely labelled cardiomyocytes within the ventricular chamber at 54 hpf by transient, mosaic expression of *myl7*:lck-EGFP in the absence of *fzd7a* and *vangl2* revealed changes in their cell morphology when compared to labelled cells within wild-type hearts (Fig. 4g–l). We observed that compared to wild-type cells (Fig. 4g, h), the basal, luminal cell surface of both *fzd7a*- and *vangl2*-deficient cells appeared disordered (Fig. 4i–l); *fzd7a* mutant cells often exhibited long cellular protrusions

(Fig. 4i, j, Supplementary Fig. 4d–f). In addition, absence of *fzd7a* led to incidental apical constriction of cardiomyocytes (Fig. 4i). These data indicate that PCP signalling coordinates cell morphology within the myocardial epithelium during cardiac chamber formation.

Next, we addressed whether the observed effects on cellular shape in the absence of PCP signalling correspond to changes in the contractile actomyosin network. Cell neighbour exchange during epithelial remodelling is characterised by junctional enrichment of both actin and myosin[27]. Phosphorylation and

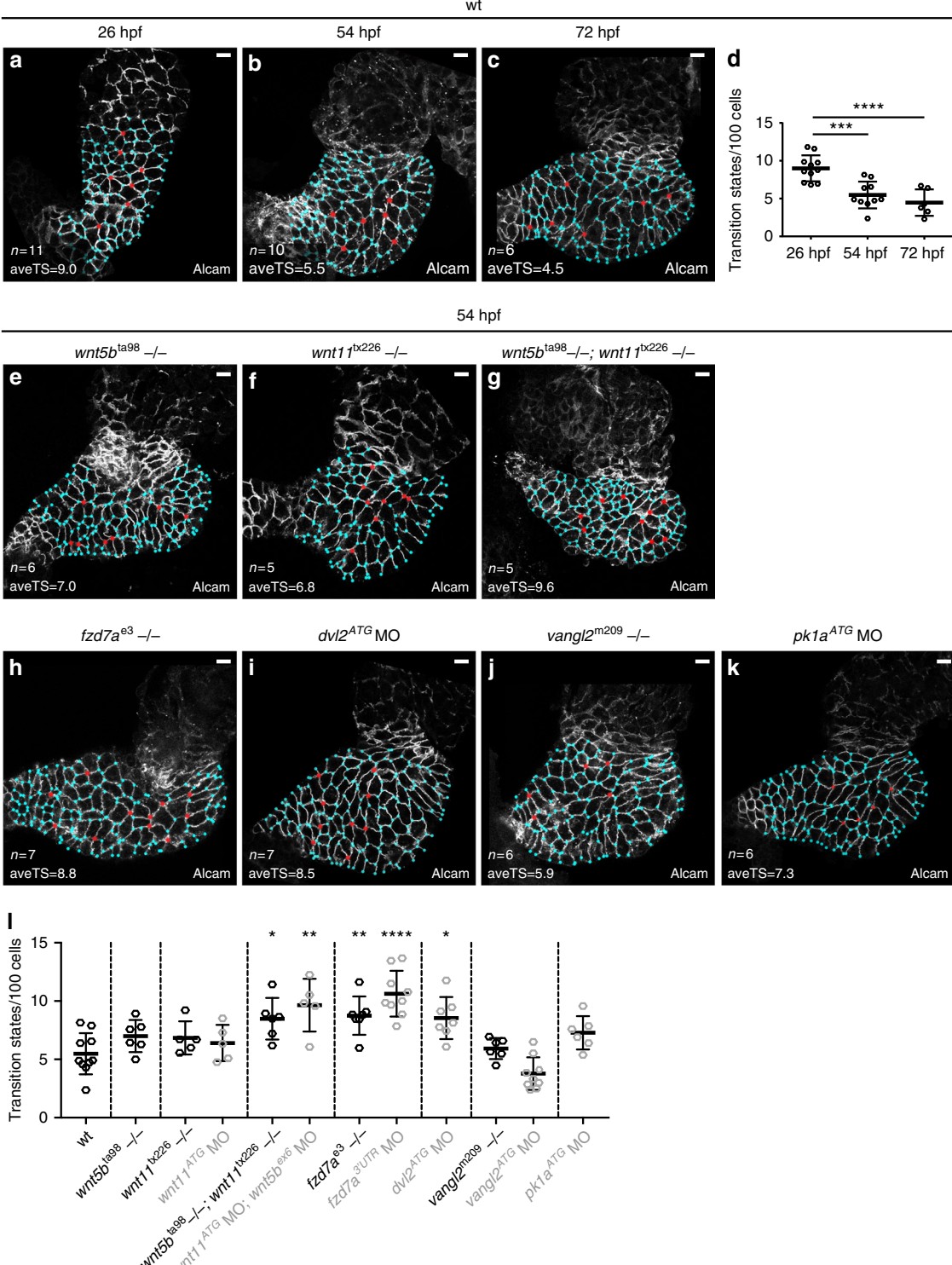

**Fig. 2** Non-canonical ligands Wnt5b and Wnt11 and PCP core components guide epithelial remodelling during chamber formation. **a–c**, **e–k** Top-down 0.5-μm confocal sections of hearts stained for Alcam (membrane marker) with the transition states in red and 3-point vertices in blue. aveTS, average number of transition states. **a–c** wt hearts at 26, 54 and 72 hpf. **d** Quantification of transition states per 100 cells in wt hearts. The LHT at 26 hpf exhibits 9.0 transitions/100 cells ($n = 11$). The two-chambered heart at 54 hpf shows 5.5 transitions/100 cells ($n = 10$), ***$P = 0.0003$, and at 72 hpf 4.5 transitions/100 cells ($n = 6$), 26 hpf vs. 72 hpf ****$P < 0.0001$, 54 hpf vs. 72 hpf $P = 0.8157$. Means ± s.d. Ordinary one-way ANOVA with Bonferroni's multiple comparison test. Reported $P$ values are multiplicity adjusted for each comparison. **e–k** Hearts at 54 hpf. Hearts of wnt5b[ta98] (**e**) and wnt11[tx226] mutants (**f**) show a slight increase in transition states. In hearts of double wnt5b[ta98];wnt11[tx226] mutants (**g**), the number of transition states increases significantly by 2-fold. The number of transition states is altered in hearts deficient in core PCP proteins (**h–k**). In hearts of fzd7a[e3] mutants (**h**), and in hearts with reduced levels of dvl2 (**i**), the number of transition states significantly increases. In hearts of vangl2[m209] mutants (**j**), and in pk1a-deficient hearts (**k**), the number of transition states mildly increases in comparison to wt hearts. **l** Quantification of average transition states/100 ventricular cardiomyocytes at 54 hpf. wt (5.5 transitions/100 cells, $n = 10$), wnt5b[ta98] (7 transitions/100 cells, $n = 6$), wnt11[tx226] (6.8 transitions/100 cells, $n = 5$), wnt11[ATG] MO (6.4 transitions/100 cells, $n = 5$), wnt5b[ta98];wnt11[tx226] (8.5 transitions/100 cells, $n = 5$, *$P = 0.0489$), wnt11[ATG] MO;wnt5b[ex6] MO (9.6 transitions/100 cells, $n = 6$, **$P = 0.001$), fzd7a[e3] (8.8 transitions/100 cells, $n = 7$, **$P = 0.0088$), fzd7a[5'UTR] MO (10.63 transitions/100 cells, $n = 9$, ****$P < 0.0001$), dvl2[ATG] MO (8.5 transitions/100 cells, $n = 7$, *$P = 0.0216$), vangl2[m209] (5.9 transitions/100 cells, $n = 6$), vangl2[ATG] MO (3.8 transitions/100 cells, $n = 9$), and pk1a[ATG] MO (7.3 transitions/100 cells, $n = 6$). Means ± s.d. Ordinary one-way ANOVA with Bonferroni's multiple comparison test. Reported $P$ values are multiplicity adjusted for each comparison. Scale bars, 10 μm

proper localisation of the regulatory light chain of non-muscle myosin II (MRLC) guides constriction of intercellular myosin cables that facilitate formation of transition states[41]. PCP signalling regulates actomyosin contractility through activation of the RhoA GTPase and subsequent Rho-associated protein kinase (ROCK)-dependent phosphorylation of MRLC as well as through the regulation of actin polymerisation in several tissues[42–46]. To monitor the possible effects of PCP on actomyosin activity during chamber remodelling, we made use of Tg(myl7:LifeAct-GFP)[s974] to visualise F-actin[47], and of the anti-phospho-MRLC antibody to label the active form of myosin. We verified the specificity of the anti-phospho-MRLC antibody by acute treatment of the dissected hearts with the ROCK inhibitor Y27623 for 1 h resulting in the decrease in phospho-MRLC levels (Supplementary Fig. 5). We went on to examine the localisation of both F-actin and phospho-MRLC at transitions states in control and in double wnt5b;wnt11, and single fzd7a, and vangl2 mutants as well as fzd7a- and vangl2-deficient hearts at the critical time of junctional remodelling at 30 hpf (Fig. 5a–h, Supplementary Fig. 4g). While we did not observe any polarised localisation of phospho-MRLC within the myocardial cells, we noticed the transition states with either presence or absence of phospho-MRLC (Fig. 5b, f). In contrast, F-actin was always present at the cell cortex, albeit more abundantly when together with phospho-MRLC (Fig. 5a, e). We quantified the percentage of co-localisation at the level of apical tight junctions, and found that in the control hearts, F-actin and phospho-MRLC co-localised 49% at the transition states within the ventricle and 67% in the OFT (Fig. 5i). In double wnt5b;wnt11 mutant hearts, F-actin and phospho-MRLC co-localised 65% in the ventricle, and 58% in the OFT. Absence of fzd7a resulted in reduced co-localisation of F-actin and phospho-MRLC at the transition states, especially in the OFT with only 30%, corroborated by 38% co-localisation in fzd7a morphants (Fig. 5i, Supplementary Fig. 4g). Conversely, loss of vangl2 led to 92 and 100% co-localisation of F-actin and phospho-MRLC in the ventricle in vangl2 mutants and morphants, respectively, while we observed no transition states in the OFT of all analysed hearts (Fig. 5i, Supplementary Fig. 4g). These data suggest that PCP signalling may regulate the efficiency of cell neighbour exchange by coordinating localised cycles of the actomyosin contractility at the transition states.

**PCP drives tissue-scale polarisation of actomyosin-based tension.** The markedly reduced co-localisation of F-actin and phospho-MRLC in the OFT in the absence of fzd7a, and the lack of transition states in the OFT upon loss of vangl2, prompted us to examine the levels of F-actin and phospho-MRLC throughout the whole ventricle. We found that in control hearts at 30 hpf, both F-actin and phospho-MRLC are planar-polarised on a tissue-scale in the myocardial epithelium, with higher levels in the distal ventricle and in the OFT than in the proximal region and near the AV when imaged at the level of apical tight junctions (Fig. 6a–d). In contrast, in the double wnt5b;wnt11, fzd7a, and vangl2 mutant hearts, as well as in fzd7a- and vangl2-deficient hearts, actomyosin was no longer polarised within the plane of the ventricle (Fig. 6e–p, Supplementary Fig. 6a–d, m–p).

Detailed examination of mid-sagittal optical sections through the single-layered myocardium revealed that this tissue-scale polarity results from apical accumulation of actomyosin in the distal ventricle/OFT region (Fig. 7a). The line plot profiles of apical (blue) and basal (orange) sides of both IC and OC (Fig. 7b, c) showed increased localisation of F-actin and more prominently accumulation of phospho-MRLC at the apical side of OC membranes with the concomitant reduction at the basal side (Fig. 7d, e, o), while the localisation of the membrane marker Alcam was equally distributed (Supplementary Fig. 7). In contrast, the line plot profiles of apical (blue) and basal (orange) sides of both IC and OC of the representative double wnt5b;wnt11, fzd7a, and vangl2 mutant hearts, as well as fzd7a- and vangl2-deficient hearts demonstrate the random distribution of actomyosin throughout the myocardium (Fig. 7f–o, Supplementary Fig. 6e–l, q–x).

Taken together, our findings demonstrate that PCP signalling orchestrates cardiac chamber remodelling by regulating the polarised distribution of actomyosin within the myocardium. The polarisation of actomyosin activity occurs in the apicobasal axis, and results in tissue-scale planar polarity where the AV/proximal region of the ventricle displays lower actomyosin activity at the apical membranes, while the distal/OFT portion of the ventricle exhibits apical actomyosin activity that may lead to higher tissue tension in this part of the heart tube.

**Cardiac looping requires PCP signalling.** Based on our data, we inferred that PCP-regulated actomyosin activity might be necessary for local cell neighbour exchange. On the other hand, such polarised distribution of actomyosin contractility might be also required for bulging of the ventricle during cardiac looping to create the convex ventricular OC curvature. To test whether locally restricted actomyosin in the OFT region might facilitate the shaping of the ventricular chambers and assist cardiac looping, we examined the effect on cardiac looping in fzd7a and vangl2 mutants as well as fzd7a- and vangl2-deficient embryos in comparison to control embryos (Fig. 8a–c, Supplementary Fig. 8a–c). We quantified the looping angle, defined as an angle between the plane of AVJ and the embryo midline axis as

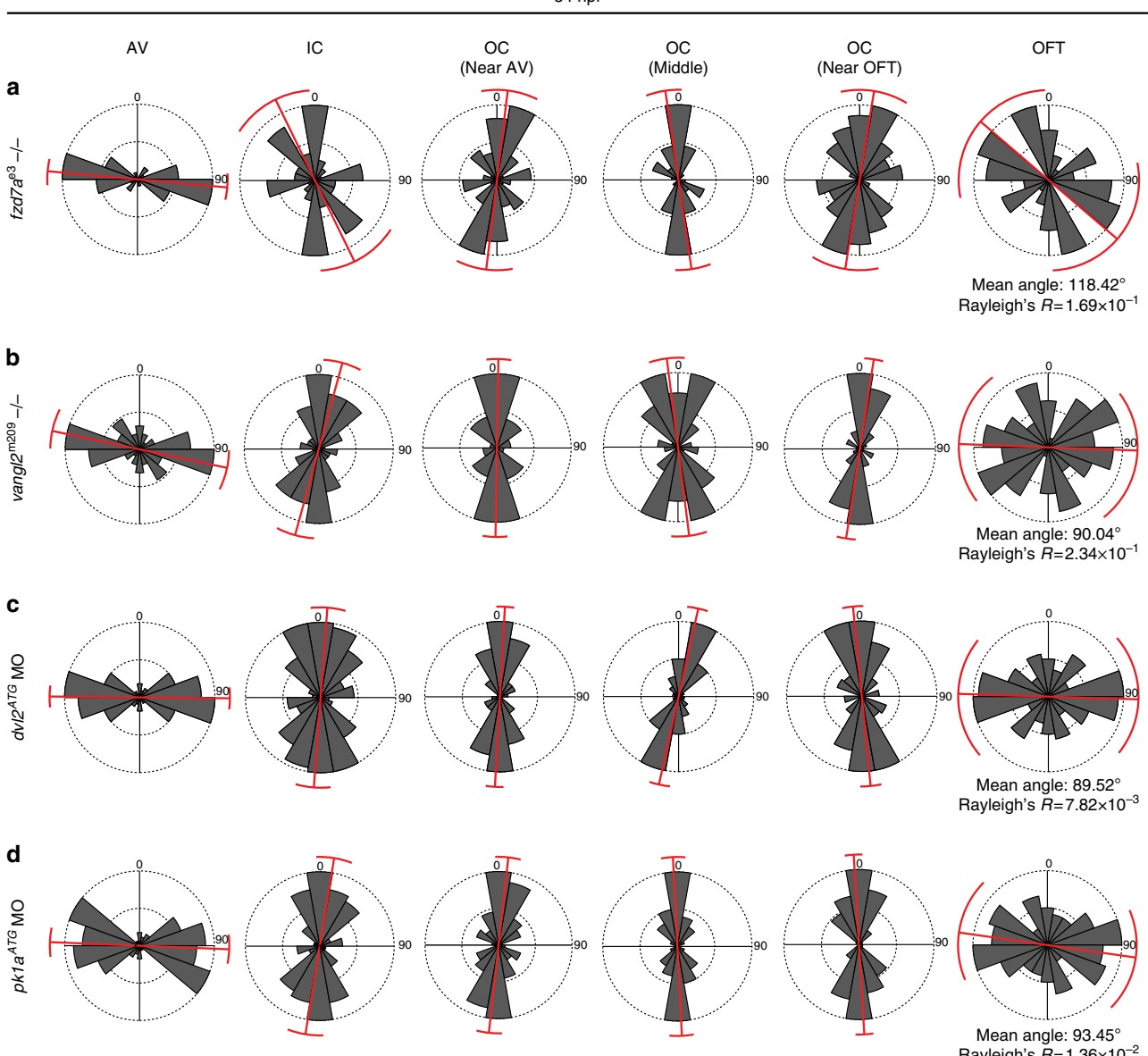

**Fig. 3** PCP regulates cardiac chamber architecture. **a** Loss of PCP signalling leads to severe defects manifested as increased variance of angle distribution (labelled in red). The main regions affected are within the OFT segment. For statistical analysis of angle distribution, Rayleigh's test assesses significance of the mean direction by using a concentration parameter of a circular distribution; the low values of Rayleigh's $R$ represent highly concentrated values with little distribution. The strongest defects in $fzd7a^{e3}$ mutant hearts are observed in the OFT segment (cells analysed, $n = 89$) with the mean angle 118.42°, and high variance Rayleigh's $R = 1.69 \times 10^{-1}$, compare to wild type (wt) in Fig. 1c, wt: mean angle OFT 92.27°, Rayleigh's $R = 9.25 \times 10^{-9}$. **b–d** Similarly, defects in the OFT cells orientation in the absence of $vangl2^{m209}$ ($n = 47$), loss of $dvl2$ ($n = 66$), and $pk1a$ ($n = 72$) are presented by higher variance of angles; $vangl2^{m209}$: mean angle OFT $= 90.04°$, Rayleigh's $R = 2.34 \times 10^{-1}$ (**b**), $dvl2^{ATG}$ MO: mean angle OFT $= 89.52°$, Rayleigh's $R = 7.82 \times 10^{-3}$ (**c**), $pk1a^{ATG}$ MO: mean angle OFT $= 93.45°$; Rayleigh's $R = 1.36 \times 10^{-2}$ (**d**)

previously described[48], to evaluate the cardiac looping efficiency in the absence of PCP signalling. We observed defective cardiac looping upon loss of both *fzd7a* and *vangl2*: while the mean looping angle in control embryos at 54 hpf is 29°, in *fzd7a*, and *vangl2* mutants as well as *fzd7a* and *vangl2* morphants the looping angle is increased to 42°, 46°, 52°, 52° respectively (Fig. 8d, Supplementary Fig. 8d). These measurements indicate that in addition to the expansion of the chambers, PCP signalling also contributes to mechanisms driving the twisting and bending of the heart tube.

Cardiac looping is tissue-intrinsic to the heart as the isolated heart tubes of teleost fish, amphibians, and chicken can bend in ex vivo culture conditions seemingly without any external cues[49–]

[51]. We hypothesised that PCP-dependent apical accumulation of phospho-MRLC in the OFT could contribute to the looping process. Consequently, hearts that lacked the specific OFT accumulation of phospho-MRLC would loose their ability to loop. To test this hypothesis, we isolated hearts from control, *fzd7a* and *vangl2* mutants as well as *fzd7a*- and *vangl2*-deficient embryo at 28 hpf from $Tg(myl7:EGFP)^{twu34}$ [52], cultured the hearts for 24 h, and analysed their looping efficiency (Fig. 8e–g). While 96% control hearts underwent looping (Fig. 8e, i), only 15% of *fzd7a* mutant (Fig. 8f, i), and 10% of *vangl2* mutant (Fig. 8g, i) hearts looped. *fzd7a*- and *vangl2*-deficient embryo failed to loop in the similar manner (Supplementary Fig. 8e–h). The looping ability ex vivo has been attributed to the actomyosin

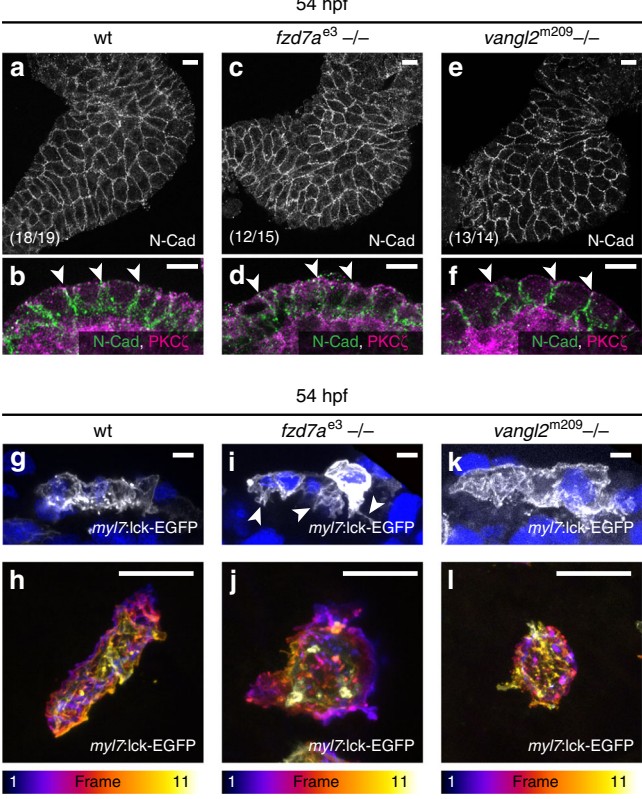

**Fig. 4** PCP affects cell morphology. **a**, **c**, **e** Top-down 1.4-μm Z-projection of 54 hpf hearts. In 18/19 wild-type (wt) hearts ($N = 3$), N-Cadherin (N-Cad) localises uniformly to cell membranes (**a**). In 12/15 fzd7a[e3] ($N = 2$) and 13/14 vangl2[m209] mutant hearts ($N = 2$) N-Cad is unchanged (**c**, **e**). **b**, **d**, **f** Mid-sagittal 1.4 μm Z-projection of the OC revealing N-Cad (green) localisation at the basolateral membranes. PKCzeta (magenta) localises to the apical tight junctions (arrowheads) in wt (**b**), fzd7a[e3] (**d**), and vangl2[m209] mutant (**f**) hearts. Scale bars, 50 μm. **g–l** OC cardiomyocytes at 54 hpf transiently expressing membrane-associated EGFP. Scale bars, 10 μm. Mid-sagittal 2.1-μm Z-projection (**g**, **i**, **k**) and maximal confocal depth Z-projection (**h**, **j**, **l**) of single OC cells depicting apical parts in blue/magenta and basal parts in white/yellow (colour scale with number of frames, z-section = 0.71 μm). At 54 hpf, wt cardiomyocytes show few basal cell protrusions (**g**, **h**; $n = 19$, $N = 3$). fzd7a[e3] mutant cardiomyocytes (**i**, **j**; $n = 12$, $N = 2$) are incidentally apically constricted, forming numerous basal cell protrusions (arrowheads in **j**). vangl2[m209] mutant cardiomyocytes are mildly affected with slightly smoother surfaces and distorted basal membrane (**k**, **l**; $n = 11$, $N = 2$). **a**, **c**, **e**, **h**, **j**, **l**; scale bars, 10 μm. **b**, **d**, **f**, **g**, **i**, **k**, **l**; scale bars, 50 μm

activity[51]. Indeed, 92% of hearts treated with 10 μM cytochalasin D, an actin polymerisation inhibitor, failed to loop as previously reported[51] (Fig. 8h, i).

Altogether, our data indicate that functional PCP signalling is required for cell neighbour exchange during cardiac chamber formation as well as for the concomitant buckling of the heart tube. Mechanistically, PCP signalling restricts the accumulation of phospho-Myosin to the apical membranes of the distal ventricle and OFT, potentially polarising the tissue tension within the heart tube.

## Discussion

Although the molecular signature governing the formation of the vertebrate cardiac chambers has been studied in great detail[3,10], the cellular processes underlying the remodelling of the LHT into

its chambered form remain enigmatic. As heart muscle cells only scarcely proliferate at the LHT stage, yet the cell number at least doubles rapidly by accrual of cardiomyocyte from the SHF, the question arises of how these cells incorporate into the heart tube and contribute to chamber expansion. The few clues to date come from the studies describing the regional cell size changes that occur during cardiac looping on the ventral side of the murine heart tube[7], the prospective zebrafish ventricular outer curvature[8], and from retrospective clonal analysis of murine heart growth[11], all suggesting that tissue remodelling might have a leading role in this process. Here, we provide evidence that the cardiac chambers form by epithelial remodelling through cell neighbour exchange guided by the PCP pathway. We propose that PCP drives cardiac chamber formation by two concomitant mechanisms: (i) by coordinating actomyosin contractility along the apicobasal axis at transition states as cells rearrange, (ii) and by planar-polarising actomyosin on a tissue-scale, thereby contributing to the cardiac looping process. Markedly, the main affected region during remodelling is the OFT, suggesting that some of the congenital heart defects associated with the deficient PCP signalling may be due to impaired cell rearrangements.

Cell neighbour exchange is a dynamic process that occurs within a range of minutes[27]. Live imaging of non-contractile hearts (Supplementary movie 1) revealed that the cell neighbour exchange indeed occurs during chamber formation, with the caveat that the timing of cell boundaries shrinking and expanding is skewed. While the cardiac chambers form and the LHT remodels, the heart already beats at around 100 times per minute, hindering the high-resolution imaging required to attain high spatial resolution data at the subcellular level. The recent advancements in the field are on track to soon address this issue[30]. As we observed the effect of PCP signalling on epithelial heart remodelling at steady state, we focused not on the dynamics of the process, but rather on its hallmark represented by the presence of the TS in the tissue. While we are unable to definitively conclude how PCP signalling regulates the resolution of the transitions states, we clearly observe that the loss of Wnt/Fzd7 signalling axis leads to marked accumulation of the TS in the tissue. The increased number of TS already at LHT in the fzd7-deficient hearts further indicates that Fzd7 may have a role already during the migration of the bilateral heart fields and/or during the LHT assembly. Nevertheless, the reduction in TS between 26 and 54 hpf in the fzd7-deficient hearts, albeit modest, suggests that Fzd7 is required also for their resolution during chamber expansion, and not only prior to the LHT formation. In contrast, the loss of Vangl2/Pk1 signalling axis has very mild effect on the accumulation of the TS in the tissue. Whether this is due to the highly dynamic nature of TS yielding unaltered net number of TS needs to be further determined. The ratio between the number of TS at 26 hpf to 54 hpf is 1.6, 1.4 and 2.1 in wild type, fzd7a- and vangl2-deficient hearts, respectively, suggesting that the TS resolution is slower in the absence of fzd7a and faster in the loss of vangl2, and warrants further examination.

During cell rearrangements, cells need to be mechanically coupled to effectively transmit force. Cadherins at cell-cell contacts mediate tissue stiffness and tissue surface tension through modulation of F-actin[53]. Although several studies have shown direct control of N-Cadherin through non-canonical Wnt signalling components in diverse contexts[54], we found that loss of fzd7a or vangl2 did not alter N-cadherin localisation or abundance at the steady state in the ventricular myocardium. We cannot, however, exclude that perturbed regulation of the PCP pathway has an effect on the rate of N-cadherin endosomal recycling as the cadherin turnover rates regulate cell rearrangements in a number of instances[55-57]. Additionally, conformational changes of cadherins at cellular junctions as well as

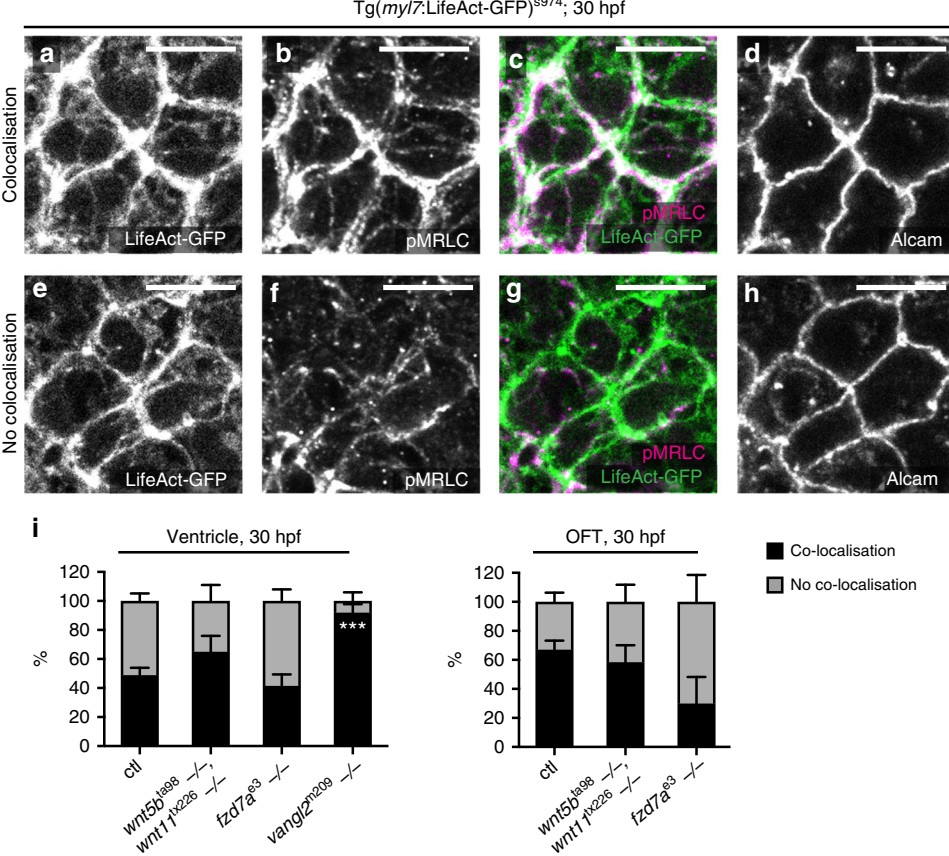

**Fig. 5** PCP affects actomyosin locally. **a**–**h** Transition states at 30 hpf stained for membrane marker Alcam (**d**, **h**), F-Actin (**a**, **e**, green in **c**, **g**) (*Tg*(*myl7*: *LifeAct-GFP*) with anti-GFP counter-staining), and pMRLC (**b**, **f**, magenta in **c**, **g**). Close-ups of exemplary T1-transition states within the ventricle of wt hearts exhibit either co-localisation of F-Actin and pMRLC (**a**–**d**), or lack of co-localisation due to the absence of pMRLC at the transition state (**e**–**h**). Scale bars, 10 μm. **i** Stacked bar graphs displaying percentage of F-Actin-pMRLC co-localisation in the ventricle and OFT at 30 hpf, respectively. In controls, 49% of F-Actin and pMRLC co-localises in the ventricle (20/41 transitions, 10 hearts); 67% co-localises in the OFT (10/15 transitions, 9 hearts). In double *wnt5b*^ta98;*wnt11*^tx226 mutants, 65% co-localises in the ventricle (16/25 transitions, 9 hearts), 58% co-localises in the OFT (4/7 transitions, 5 hearts). In *fzd7a*^e3 mutant hearts, 41% co-localises in the ventricle (13/31 transitions, 9 hearts), only 30% co-localises in the OFT (4/13 transitions, 6 hearts). In *vangl2*^m209 mutant hearts, 92% co-localises in the ventricle (39/43 transitions, 18 hearts), no transitions are observed in OFT. \*\*\*$P < 0.0003$. Means ± s.d. Ordinary two-way ANOVA with Tukey's multiple comparison test. Reported $P$ values are multiplicity adjusted for each comparison

availability of $Ca^{2+}$ affect their stability resulting in changes in the strength of intercellular adhesive bonds[58]. In this regard, the recently identified Wnt11/L-type $Ca^{2+}$ channel pathway could act on adhesion processes through its effects on intercellular $Ca^{2+}$ concentrations[59].

In the heart, coordinated contractility by actomyosin is fundamental to proper organ function. Non-muscle myosin II (NMII), a member of the myosin II subfamily that includes cardiac, skeletal, and smooth muscle myosin[60], was shown to be the key mediator of cell neighbour exchange[27–29]. NMII activity is regulated by dynamic phosphorylation of its regulatory light chain involving a tightly regulated interplay between ROCK, myosin light chain kinase (MLCK), and myosin phosphatase (MYPT)[60]. Phospho-myosin becomes enriched and localised to temporary transition states, where it appears to facilitate cell junction plasticity[27,29]. PCP signalling is known to regulate the organisation of cytoskeletal components through its effects on Rho GTPases[41,42,44,45]. Here, we demonstrate that PCP regulates the localisation of the active phosphorylated form of MRLC locally at transition states as well as on the tissue-scale at the apical membrane of distal ventricular and OFT myocardium. It is tempting to speculate that PCP may either locally tether the phospho-MRLC form, or more likely as previously suggested[61], control the local activity of the kinases that mediate the

phosphorylation of myosin regulatory light chain or their interacting partners at the apical membranes of the polarised myocardial epithelium. Alternatively, PCP may affect the localisation of actomyosin through regulation of apicobasal polarity itself, for instance through binding of Dvl to aPKC[62].

Polarised actomyosin contractility is an essential driving force underlying epithelial remodelling[61]. Reportedly, NMII localises asymmetrically in the cardiac fields prior to their convergence and formation of the LHT[63]. It has been suggested that such asymmetric actomyosin activity facilitates tissue-intrinsic dextral cardiac looping independent of asymmetry-breaking signals, such as Nodal[51]. Mechanisms that might mediate polarised tissue tension in the myocardial epithelium only begin to emerge. The recently reported oriented epithelial tension in the SHF cells of the dorsal pericardial wall in mice[64] may be transmitted to the myocytes of the growing heart tube by cell flow, a process similar to wing elongation in Drosophila[65]. Here, we demonstrate that in addition to the tissue-intrinsic and self-organising properties of the heart tube and adjacent SHF, PCP pathway provides instructive cues to planar polarise actomyosin contractility in the distal ventricle and the OFT, resulting in polarised tissue tension that is required to facilitate cardiac looping and bulging of the cardiac chambers out of the LHT. This is in accordance with the recent findings showing that multicellular gradient of the myosin

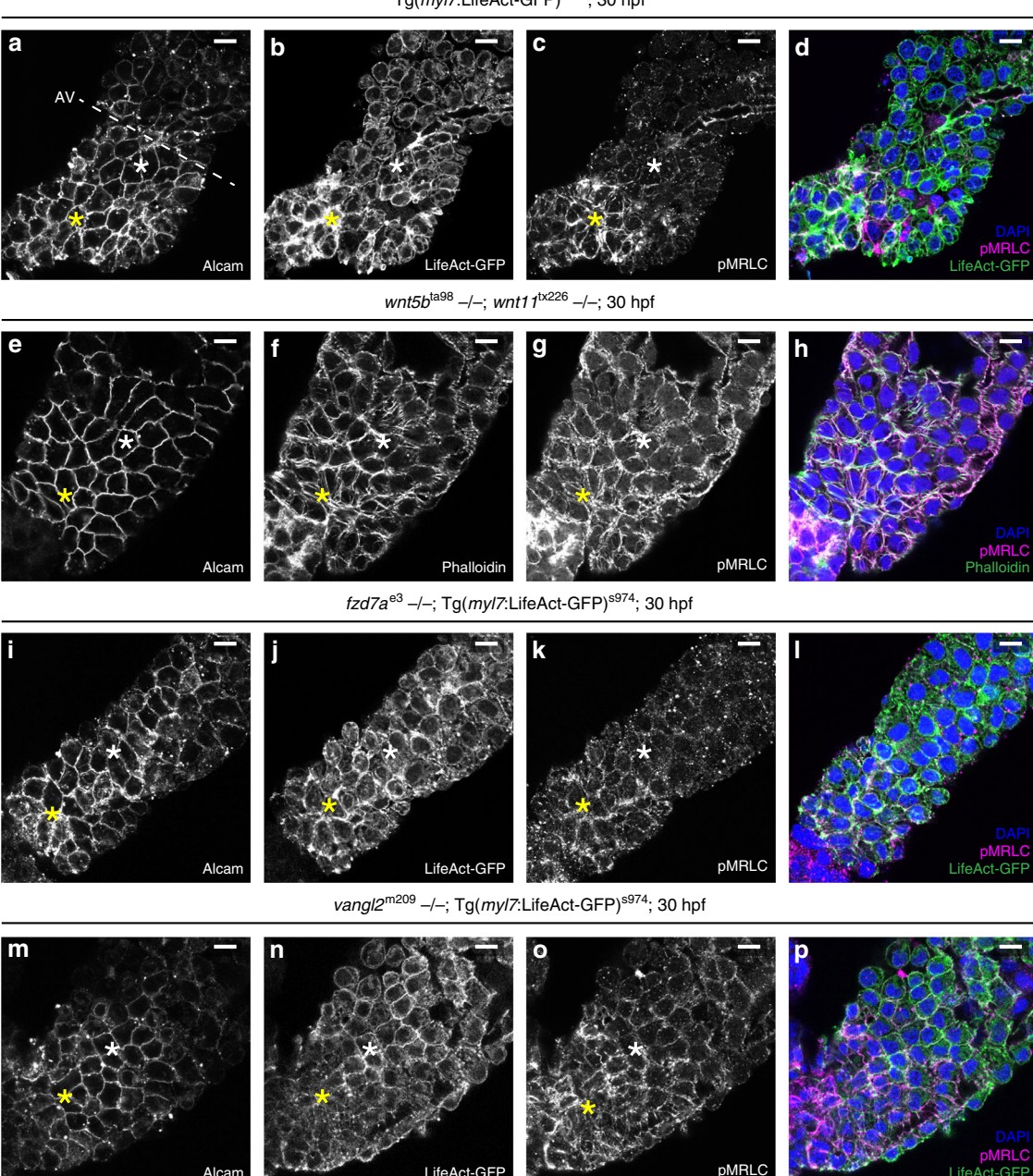

**Fig. 6** Tissue-scale planar polarisation of actomyosin. **a–p** Top-down 1-µm Z-projection of the linear heart tube at 30 hpf stained for Alcam (**a**, **e**, **i**, **m**) to visualise ventricular membranes, (*Tg*(*myl7*:LifeAct-GFP, counter-stained with anti-GFP) (**b**, **j**, **n**) or Phalloidin (**f**) to visualise F-actin (green in **d**, **h**, **l**, **p**), and pMRLC (**c**, **g**, **k**, **o**, magenta in **d**, **h**, **l**, **p**). The proximal ventricular region is labelled with a white asterisk, the distal ventricular region with a yellow asterisk, (**a–c**, **e–g**, **i–k**, **m–o**). In control hearts both F-actin (**b**, green in **d**) and pMRLC (**c**, magenta in **d**) are tissue-scaled planar-polarised in the myocardial epithelium, with higher levels in the distal ventricular region (yellow asterisk) and the OFT than in the proximal ventricular region (white asterisk) and near the AV (8/10 hearts with planar-polarised actomyosin, N = 2). In 30 hpf linear heart tubes of double *wnt5b*[ta98];*wnt11*[tx226] mutants (**e–h**, 0/8 hearts with planar-polarised actomyosin, N = 2), *fzd7a*[e3] mutants (**i–l**, 1/9 hearts with planar-polarised actomyosin, N = 2), and *vangl2*[m209] mutants (**m–p**, 3/15 hearts with planar-polarised actomyosin, N = 2), both F-actin (**f**, **j**, **n**, green in **h**, **l**, **p**) as well as pMRLC (**g**, **k**, **o**, magenta in **h**, **l**, **p**) are localised throughout the ventricle. Hearts were counter-stained for DAPI to label nuclei (blue in **d**, **h**, **l**, **p**). Scale bars, 10 µm. AV, atrio-ventricular junction

activity is essential for bending and folding of the ventral epithelium during *Drosophila* gastrulation[66]. Additionally, the recent study of cardiac looping in mice demonstrated requirement for asymmetries generated at mechanically constraint poles of the heart tube that contribute to buckling mechanism[67].

In addition to the aforementioned myocardial tissue-scale forces, hemodynamic forces, interaction with the endocardium or biomechanical signalling contribute to the heart morphogenesis

as well-documented mutants lacking endocardium or a functional contractile apparatus fail to loop and to form proper cardiac chambers[31,68]. The heart is the first functional organ, and morphogenetic processes that drive cardiac chamber formation occur concomitantly with the establishment and refinement of cell coupling. Thus, multiple genetic pathways as well as physical cues, including mechanical forces, need to converge to establish the final cardiac form and function.

Altogether, our work establishes that Wnt non-canonical PCP signalling coordinates cardiac chamber remodelling through tissue-scale polarisation of actomyosin. Our findings underline the importance of understanding the role of PCP during heart development, as congenital heart defects that frequently associate with defects in OFT remodelling, are attributed to loss of non-canonical Wnt ligands or to mutations in core components of the PCP pathway[26]. Elucidating these mechanisms is therefore pivotal not only for understanding the fundamental principles of heart development, but also for improving treatment options of both congenital and acquired heart diseases.

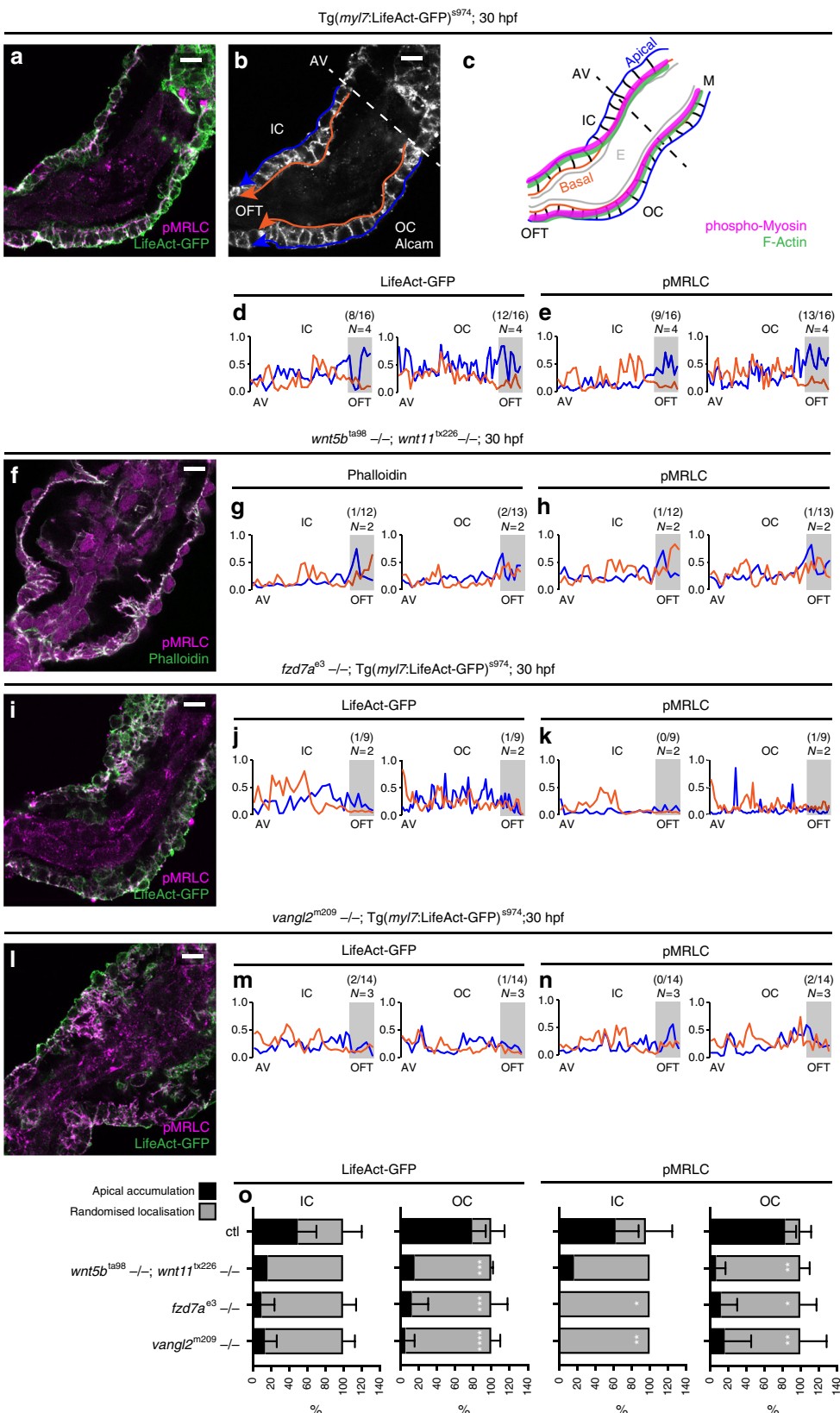

**Fig. 7** Loss of planar-polarised actomyosin in the absence of PCP signalling. **a, f, i, l** Mid-sagittal 1-μm *Z*-section through 30 hpf LHT of ctl (**a**), *wnt5b*[ta98]; *wnt11*[tx226] (**f**), *fzd7a*[e3] (**i**), and *vangl2*[m209] mutants (**l**). **a** In controls, F-actin (*Tg*(*myl7:LifeAct-GFP*) with anti-GFP counter-staining) and pMRLC accumulate apically in the distal/OFT region, Alcam visualises cell membranes (**b**). **c** Schematic of a mid-sagittal section through the heart tube with apical myocardial membrane in blue, basal membrane in orange, actin in green and phospho-myosin in magenta. **d, e, g, h, j, k, m, n** Line plot profiles of apical (blue) and basal (orange) membranes (as in **b, c**, arrows in **b** indicate the direction of the line plots), OFT region highlighted in grey. **d, e** In controls, line plot profiles of IC and OC reveal apical accumulation (aa) in the OFT and randomised localisation in proximal ventricle of both F-actin (**d**) and pMRLC (**e**), depicted in **c** with green and magenta lines, respectively. Numbers represent the number of hearts showing the presence of aa out of total number of hearts. **g, h, j, k, m, n** Line plot profiles through the hearts of *wnt5b*[ta98];*wnt11*[tx226] (**g, h**), *fzd7a*[e3] (**j, k**), and *vangl2*[m209] (**m, n**) mutants revealing randomised actomyosin localisation. **o** Quantification of actomyosin localisation in IC and OC of 30 hpf hearts of *wnt5b*[ta98];*wnt11*[tx226], *fzd7a*[e3], and *vangl2*[m209] mutants demonstrate lower aa in the OFT of both F-Actin and pMRLC as compared to ctl. aa of F-actin in IC: ctl (50%); *wnt5b*[ta98];*wnt11*[tx226] (17%); *fzd7a*[e3] (10%); *vangl2*[m209] (14%). aa of pMRLC in IC: ctl (63%); *wnt5b*[ta98];*wnt11*[tx226] (17%); ctl vs. *fzd7a*[e3] (0%), *$P = 0.0126$; ctl vs. *vangl2*[m209] (0%), **$P = 0.0046$. aa of F-actin in OC: ctl (79%) vs. *wnt5b*[ta98];*wnt11*[tx226] (15%), ***$P = 0.0009$; ctl vs. *fzd7a*[e3] (13%), ***$P = 0.0006$; ctl vs. *vangl2*[m209] (6%), ****$P < 0.0001$. aa of pMRLC in OC: ctl (83%) vs. *wnt5b*[ta98];*wnt11*[tx226] (7%), **$P = 0.0066$; ctl vs. *fzd7a*[e3] (13%), *$P = 0.0119$; ctl vs. *vangl2*[m209] (17%), **$P = 0.007$. Means ± s.d. Ordinary two-way ANOVA with Tukey's multiple comparison test. Scale bars, 10 μm

## Methods

**Zebrafish**. Zebrafish were bred, raised, and maintained in accordance with the guidelines of the Max-Delbrück Center for Molecular Medicine and the local authority for animal protection (Landesamt für Gesundheit und Soziales, Berlin, Germany) for the use of laboratory animals, and followed the 'Principles of Laboratory Animal Care' (NIH publication no. 86-23, revised 1985) as well as the current version of German Law on the Protection of Animals. Zebrafish strains *AB*, *TüLF,* and *Wik* were used for analysis of wild-type phenotypes and for injection of constructs and morpholinos. Mutant lines used in this study included *fzd7a*[e3] [37], *vangl2*[m209] [38], *wnt11*[tx226] [36], and *wnt5b*[ta98] [35]. Embryos were kept in E3 embryo medium (5 mM NaCl, 0.17 mM KCl, 0.33 mM CaCl$_2$, 0.33 mM MgSO$_4$, pH 7.4) under standard laboratory conditions at 28.5 °C. Staging was performed as described previously by hours post fertilisation (hpf) or by counting somites. *Tg* (*myl7:EGFP*)[twu34], *Tg*(*myl7:LifeAct-GFP*)[s974], and *Tg*(*myl7:lck-EGFP*)[md71] lines were used as published[30,47,52].

**Morpholinos**. The following morpholinos obtained by Gene Tools, LLC were used: *MO1-dvl2* (5′-TAAATTATCTTGGTCTCCGCCATGT-3′) (ZFIN ID: ZDB-MRP HLNO-060724-1), *MO1-fzd7a* (5′-ATAAACCAACAAAAACCTCCTCGTC-3′) (ZFIN ID: ZDB-MRPHLNO-050923-5), *MO1-pk1a* (5′-GCCCACCGTGATTCT CCAGCTCCAT-3′) (ZFIN ID: ZDB-MRPHLNO-060209-7), *MO1-tnnt2a* (5′-CATGTTTGCTCTGATCTGACACGCA-3′) (ZFIN ID: ZDB-MRPHLNO-060317-4), *MO1-vangl2* (5′-GTACTGCGACTCGTTATCCATGTC-3′) (ZFIN ID: ZDB-MRPHLNO-041217-5), *MO2-wnt11* (5′-GTTCCTGTATTCTGTCATGTCGCTC-3′) (ZFIN ID: ZDB-MRPHLNO-050318-3), *MO2-wnt5b* (5′-TGTTTATTTCCTCACC ATTCTTCCG-3′) (ZFIN ID: ZDB-MRPHLNO-051207-1). Except *MO2-wnt5b* that targets the *3′* end of the exon–intron junction of exon 3, all morpholinos used are blocking translation.

**Time-lapse in vivo imaging**. Embryos were anaesthetised with 0.016% tricaine (w/ v) and embedded in a ventral position in 1% low melting agarose for immobilisation in glass bottom MatTek dishes. In vivo imaging was performed using a Leica SP8 microscope with a ×25 (0.95 numerical aperture) HCX IRAPO water immersion objective. During acquisition, embedded zebrafish were kept in a chamber with 28 °C and covered with E3 medium containing 0.016% tricaine.

**Heart-looping measurements**. To measure the angle of cardiac looping, *Tg*(*myl7: EGFP*)[twu34] or *Tg*(*myl7:LifeAct-GFP*)[s974] embryos at 54 hpf were embedded in 1.5% methylcellulose (Sigma) dissolved in E3 embryo medium, and imaged with the Leica M80 stereomicroscope. Images were analysed using ImageJ/Fiji.

**Quantification of transition states followed by analysis of cell shape and orientation**. Measurement of cell area, perimeter, elongation, and polarity analysis is carried out using Packing Analyzer v2.0[65]. For cell orientation analysis, a simple local anatomical model is fitted to each heart in MatLab (Mathworks); first the ventricular circumference is manually traced on each heart, then smoothed using Bezier curves, and the cardiac center is defined as the centroid of the smoothed circumference points (Fig. 1a). Four clearly defined surface anatomical landmarks (1–4, numbers labelled in blue in Fig. 1a) are manually selected and are used for cardiac segmentation: 1: where atrio-ventricular-junction (AV-junction) joins the inner-curvature, 2: where inner-curvature joins the ventricular-outflow, 3: where the ventricular-outflow joins the outer-curvature, 4: where the outer-curvature joins the AV-junction. Two additional points are calculated: one-third (5) and two-thirds distance (6) along the outer-curvature circumference. These six points, along with the ventricular centroid, define six cardiac segments: IC (lesser-curvature), OFT (ventricular-outflow), OC near OFT (greater-curvature-I), OC middle (greater-curvature-II), OC close to AV (greater-curvature-III) and AV. The angle of orientation of the cell to the cardiac surface normal was determined for each peripheral cell. This angle is on the scale ±90° and is defined as the angle between

the cell orientation vector (Ψ, indicated by a red line in Fig. 1a) and the normal to the local ventricular surface tangent (the normal is shown as a black line and the tangent as a green line in Fig. 1a). The angle on the scale ±90° between the cell orientation vector (Ψ) and the normal to the local ventricular surface tangent is determined for each cell on the ventricular circumference.

**Heart explants**. The experiment was performed as reported previously with minor changes[51]. Briefly, at 28 hpf, hearts were manually dissected from *Tg*(*myl7:EGFP*)[twu34] either uninjected controls or injected with *fzd7a*[5′UTR] or *vangl2*[ATG] MO or *fzd7a*[e3] mutants and *vangl2*[m209] mutants, and placed into supplemented L-15 culture medium (15% fetal bovine serum, 0.8 mM CaCl$_2$, 1:200 Penicillin-Streptomycin (ThermoFisher Scientific™)) in Leibovitz's L-15 Medium, Gluta-MAX™ Supplement (ThermoFisher Scientific™). Explants were incubated at 28.5 °C for 24 h and fixed with Shandon™ Glyo-Fixx™ (Cat#9990920; ThermoFisher Scientific™), or 4% Formaldehyde in PEM buffer (0.1 M PIPES (pH 6.95), 2 mM EGTA, 1 mM MgSO$_4$) with 0.1% Triton to avoid quenching of fluorescence, for 20 min at RT. Immunostaining and image analysis was performed as described below.

**Immunostaining and confocal microscopy followed by image analysis**. Hearts were dissected from 26, 30, 54, 72 hpf zebrafish embryos in normal Tyrode's solution (NTS) (136 mM NaCl, 5.4 mM KCl, 1 mM MgCl$_2$ × 6H$_2$O, 5 mM D(+) Glucose, 10 mM HEPES, 0.3 mM Na$_2$HPO$_4$ × 2H$_2$O, 1.8 mM CaCl$_2$ × 2H$_2$O; pH 7.4) with 20 mg mL$^{-1}$ BSA and fixed with Shandon™ Glyo-Fixx™ or 4% Formaldehyde in PEM buffer for 20 min at RT. The hearts were incubated in blocking solution (BS) (PBS; 5% Normal Goat Serum; 1% DMSO, 0.1% Tween-20, 2 mg mL$^{-1}$ BSA) for at least 2 h, and then stained with the primary antibodies diluted in BS over night at 4 °C: mouse anti-zn8 (Alcam) (Developmental Studies Hybridoma Bank; RRID: AB_531904) 1:50, chicken anti-GFP (Cat#GFP-1010; Aves Lab; RRID: AB_2307313) 1:100, rabbit anti-GFP (Cat#ab209, abcam; RRID: AB_303395) 1:100, rabbit anti-phospho-Myosin (S20) cardiac (pMyl9/pMyl12) (Cat#ab2480; abcam; RRID:AB_303094) 1:100, mouse anti-N-Cadherin (Cat#610920; BD Transduction Laboratories™; RRID: AB_2077527) 1:100, rabbit anti-PKC-zeta (Cat#sc-216, Santa Cruz Biotechnology, RRID:AB_2300359) 1:100. After three washing steps in BS for 30 min hearts were incubated in secondary antibodies (diluted 1:500 in BS; Phalloidin diluted 1:4 in BS) for at least 2 h at RT: Alexa Fluor™ 488 Phalloidin (Cat#A12379, ThermoFisher Scientific™), Goat anti-Chicken IgY conjugated with FITC (Cat#F-1005; Aves Lab), Goat anti-Mouse IgG conjugated with Alexa Fluor 633 (Cat#A-21052; Life Technologies), Goat anti-Rabbit IgG (H + L) conjugated with Alexa Fluor 555 (Cat#A-21428; Life Technologies), and mounted after over night washing in BS in the ProLong Gold antifade reagent with 4,6-diamidino-2-phenylindole (Cat#P36935; ThermoFisher Scientific™). Confocal images were obtained using the Leica SP5 with a ×63 oil immersion objective and processed using ImageJ/Fiji, Packing analyzer v2.0, and Photoshop.

**Drug treatments**. Heart explants dissected from 28 hpf *Tg*(*myl7:EGFP*)[twu34] embryos were treated with 10 μM Cytochalasin D (Sigma) in supplemented L-15 culture medium for 24 h at 28.5 °C. For ROCK-inhibition dissected hearts at 54 hpf were treated for 1 h with 200 μM Y-27632 dihydrochloride (abcam, dissolved in DMSO) in NTS at RT prior to fixation with Shandon™ Glyo-Fixx™ for 20 min at RT. Treated hearts were then incubated in BS for at least 2 h and processed for immunostaining as outlined.

**Fluorescence intensity measurements**. Line scans (line width: 10) to determine fluorescence intensity profiles were analysed using ImageJ/Fiji. Using the Alcam staining a line along the apical and basal side of the single-layered myocardium at 30 hpf was defined from the AV to the OFT of the ventricle. The pixel intensities were averaged on a line width of 10 pixels to reduce noise. The pixel intensity (grey values, 8 bit) was plotted against the distance (micrometres) along the defined line for pMRLC and myl7:LifeAct-EGFP stainings. The intensity values were

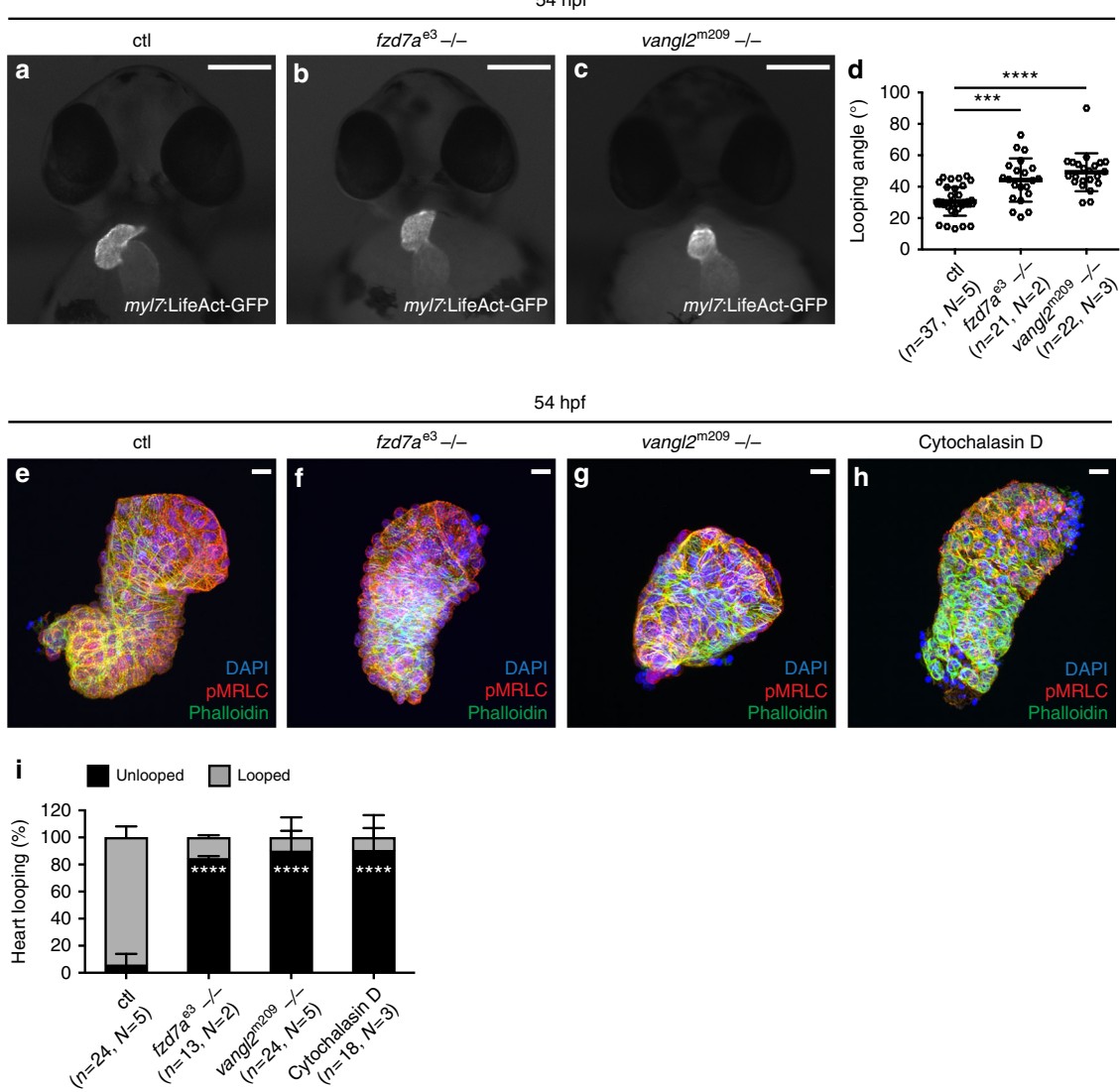

**Fig. 8** Cardiac looping requires PCP signalling. **a**–**c** Cardiac looping of *Tg*(*myl7*:*LifeAct-GFP*) embryos at 54 hpf. Scale bars, 50 μm. Cardiac looping in *fzd7a*[e3]
(**b**), and *vangl2*[m209] mutant (**c**) hearts is impaired as compared to control (**a**) hearts. **d** Quantification of cardiac looping angle, defined as an angle between
the plane of atrio-ventricular junction and the embryo midline axis. *fzd7a*[e3] (**b**) and *vangl2*[m209] (**c**) mutant hearts display significantly greater average
looping angle than control. Means ± s.d. Ctl vs. *fzd7a*[e3]−/−, ***$P = 0.0001$; ctl vs. *vangl2*[m209]−/−, ****$P < 0.0001$ using ordinary one-way ANOVA with
Bonferroni's multiple comparison test. Reported *P* values are multiplicity adjusted for each comparison. **e**–**h** Cardiac looping of explanted hearts from ctl
(**e**), *fzd7a*[e3] (**f**) or *vangl2*[m209] (**g**) mutant embryos, and hearts treated with 10 μM cytochalasin D (**h**). At 28 hpf linear heart tubes were dissected,
incubated in supplemented tissue culture medium for 24 h prior to fixation, stained with DAPI (**e**–**h**), pMRLC (**e**–**h**), and Phalloidin (**e**–**h**), and imaged.
cytochalasin D-treated hearts do not loop (**h**), similarly to the absence of *fzd7a* and *vangl2*. Scale bars, 10 μm. **i** Quantification of cardiac looping of
explanted hearts reveal the significant increase of unlooped heart explants from *fzd7a*[e3] or *vangl2*[m209] mutants, and upon cytochalasin D treatment in
comparison to ctl. Ctl (6% unlooped, 2/24, N = 5) vs. *fzd7a*[e3]−/− (85% unlooped, 11/13, N = 2), ****$P = 0.0001$; ctl (unlooped) vs. *vangl2*[m209]−/− (90%
unlooped, 22/24, N = 5), ****$P = 0.0001$; ctl (unlooped) vs. cytochalasin D treatment (90% unlooped, 16/18, N = 3), ****$P = 0.0001$. Means ± s.d.
Ordinary two-way ANOVA with Dunnett's multiple comparison test. Reported *P* values are multiplicity adjusted for each comparison

normalised to the lowest and highest intensity within the selection and plotted with
Microsoft Excel. To quantify ventricular fluorescence intensities, the mean intensity
is defined by selection of the ventricular area from top-down confocal sections in
ImageJ/Fiji. Substraction of the background is followed by adjustment towards the
organ size by division with each individual size of the ventricle. Normalisation is
conducted by division by the total average.

**Statistics**. Hearts, in which immunostaining failed or which were damaged were
excluded from the samples. Appropriate statistical tests were used for each sample.
No randomisation was used, blinding was used. Statistical analysis and testing on
cell shape is carried out using the Circular Statistics Toolbox[69] for MatLab
(Mathworks), custom written code, and the PAST statistics package[70]. As a test of
uniform distribution the Rayleigh's *R* statistic was used[71] and is applied for the

analysis of axial data using the method proposed by Davis et al.[72] Statistical ana-
lysis of data concerning number of transition states/100 cells and cardiac looping
angle (°) were performed using ordinary one-way ANOVA with Bonferroni's
multiple comparison test. Statistical analysis of data regarding co-localisation of F-
Actin and pMRLC at transition states and apical accumulation of actomyosin in
mid-sagittal sections were performed using ordinary two-way ANOVA with
Tukey's multiple comparison test. Ordinary two-way ANOVA with Dunnett's
multiple comparison test was used to quantify cardiac looping of explanted hearts.
Reported *P* values were all multiplicity adjusted for each comparison. Statistical
analysis of data regarding cell area in μm$^2$, circularity, and ventricular fluorescence
intensities of pMRLC were performed using unpaired *t*-test with Welch correction,
reported *P* values are two-tailed. Capital *N* represents number of independent
biological experiments in Figures and corresponding legends. Uncapitalised *n*

describes number of cardiomyocytes in Fig. 1b and in the legend for Fig. 3 and number of hearts in Fig. 2 as well as in Fig. 4a, c and e. In Fig. 8d, uncapitalised *n* stands for the number of embryos. In Fig. 8i, uncapitalised *n* represents the number of explanted hearts.

**Code availability**. The custom MatLab script 'Zf heart cell orientation.rar' used to analyse the cell orientation is available as Supplementary Software 1.

**Data availability**. The data that support the findings in this study are available within this article and its Supplementary Information files, and from the corresponding author upon reasonable request.

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

## Acknowledgements

We thank C.-P. Heisenberg for *fzd7a*^e3^−/−, M. Wiweger for *vangl2*^m209^−/−, and D. Stainier for *Tg(myl7:LifeAct-GFP)*^s974^ fish strains; I. Fechner, J. Richter, C. Schulz, and R. YanDo for technical support. We thank Oliver Rocks and Mariana Guedes Simões for comments, and teams of Advanced Light Microscopy Facility and Fish Facility at MDC for expert support. This work has been supported by the Helmholtz Young Investigator Program VH-NG-736, (DFG) PA2619/1-1, and Marie Curie Career Integration Grant from the European Commission (MC CIG) (WNT/CALCIUM IN HEART-322189) to D. P., by the excellence cluster REBIRTH, SFB958 and DFG SE2016/10-1 to S.A.-S, by the Canton of Zürich, a Swiss National Science Foundation (SNSF) professorship (PP00P3_139093), MC CIG (PCIG14-GA-2013-631984), and a Swiss Heart Foundation grant to C.M.

## Author contributions

A.M.Merks, M.S., A.M.M, N.V.M., I.O., S.D., A.B. and D.P. performed experiments, and analysed the data. M.S., A.M.Merks, S.A.-S., C.M., and D.P. designed research. S.G. wrote cell shape and orientation analysis. D.P. wrote the manuscript with support from M.S., A. M.Merks, S.A.-S., and C.M.
