## [Peer Review File · Nature Communications]

Reviewers' Comments:

Reviewer #1:

Remarks to the Author:

This paper by the group of Daniela Panakova addresses the role of Planar Cell Polarity (PCP) signaling during the remodeling of the heart tube in zebrafish. The authors analyze the morphogenetic cellular re-arrangement that take place between the liner heart tube stage and the mature heart and how the Wnt5/11 and core Frizzled (Fz)-PCP factors influence the process. Overall the data are of high quality and are presented adequately. With some revisions (see below) the paper is worth considering for Nature Communications and will make an exciting contribution to the field and to PCP-regulated morphogenesis processes in general

There are a few issues with the paper, as currently presented, that should be addressed before the paper is published.

Specific comments:

i) While the data presentation overall is adequate, there are some instances where better data should be added. For example, all analyses of Vangl2 in figures 6 and 7 rely on morphant experiments, there are Vangl2 mutant alleles/null alleles (used in this study and otherwise) and these are better tools.

ii) The presentation of the data figures could be arranged differently to allow direct comparison of wild-type vs the individual mutant scenarios. As it is now Figure 1 is all wild-type while figures 2-3 is all mutant. It would be better to align wild-type with mutant for direct comparison and split the figure panels into figures along the assays or measurement types used in analyses.

Similarly, Figures 5-6 should be merged to allow direct comparison (with possibly some aspects going to a Supplemental Figure).

iii) In general terms, the text is very descriptive with many, many details stated in the main text, including many percentages of effects and associated statistical analyses, and then again repeated in the figure legends and also shown in the figures. This applies to the whole Results section, for example lines 128-133 or lines 141-148, or 1154-160, or 169-172, or 180-183. It is very difficult to read with all the numerical values inserted in the main text.

iv) In the Introduction on PCP, lines 65-72, the authors just list one review from Yingzi Yang as reference. While that review is fine, the authors should remember that all the PCP knowledge there is, it comes from work in *Drosophila* (without which the authors would not be able to write the paper...). As such they should recognize the *Drosophila* work as basis for their own and cite a few recent reviews from the key *Drosophila* labs in the field. Examples to cite would be:

- Singh J, and Mlodzik M. (2012). Planar cell polarity signaling: coordination of cellular orientation across tissues. *Wiley Interdiscip Rev Dev Biol.* 2012 Jul-Aug; 1(4):479-99.
- Goodrich LV, and Strutt D. (2011). Principles of planar polarity in animal development. *Development.* 2011 May; 138(10):1877-92.
- Adler PN. (2012). The frizzled/stan pathway and planar cell polarity in the *Drosophila* wing. *Curr Top Dev Biol.* 2012; 101:1-31.
- Fanto M, McNeill H. (2004). Planar polarity from flies to vertebrates. *J Cell Sci.* 2004 Feb 1; 117:527-33.
- Bayly R, Axelrod JD. (2011). Pointing in the right direction: new developments in the field of planar cell polarity. *Nat Rev Genet.* 2011 Jun; 12(6):385-91.

v) Pge 9 Lines 212-214: The original work showing that Rho-family GTPases are involved in PCP signaling should be included:

- Eaton S, Wepf R, and Simons K. (1996). Roles for Rac1 and Cdc42 in planar polarization and hair outgrowth in the wing of *Drosophila*. *J Cell Biol.* 1996 Dec; 135(5):1277-89.
- Strutt DI, Weber U, and Mlodzik M. (1997). The role of RhoA in tissue polarity and Frizzled

signalling. Nature. 1997 May 15;387(6630):292-5.

- Boutros M, Paricio N, Strutt DI, and Mlodzik M. (1998). Dishevelled activates JNK and discriminates between JNK pathways in planar polarity and wingless signaling. Cell. 1998 Jul 10;94(1):109-18.

Reviewer #2:

Remarks to the Author:

This paper deals with the interesting process of cardiac chamber ballooning, and more specifically with the behavior of the cardiomyocytes during this process. The authors put forth a model by which the Planar Cell Polarity (PCP) pathway coordinates the localization of actomyosin activity and thus the efficiency of cell neighbor exchanges. Further claims include tissue tension in the control of cardiac chamber ballooning as well as cardiac looping. While the questions addressed are important, there are a number of important issues with the current manuscript that need to be addressed in full before it can be considered further.

General comments:

1. The cell-transition experiments were done in non-contracting hearts; not clear how physiologically relevant these data are considering that these rearrangements will be strikingly different in the beating heart (as acknowledged in the MS). In other words, the experiments will need to be done again in contracting hearts.

2. The authors equate Wnt5b and Wnt11 with PCP signaling, a now discounted simplification (<http://web.stanford.edu/group/nusselab/cgi-bin/wnt/receptors>)

3. morphants and mutants: some of the phenotypes are fairly subtle (and variable), thus one should take advantage of the blinding aspect of a genetic approach to double check claims (ie, cross two hets together, analyze the embryos, sort them according to phenotype and then genotype).

4. The manuscript is not easy to read, some images are of low quality and/or insufficient magnification, some key references appear to be missing, and the work would benefit from using some recently published reagents (including transgenic lines).

Other comments:

Figure 1a, 1b and 1c

Please show the different cell shapes at IC and OC and cell orientation.

Based on the drawing, the orientation (Fig 1b) of the CMs in the AV canal is not clear; transition states and 3-point vertices were not precisely marked in all conditions.

check effect on transition states in wnt5b, wnt11, fzd7a and vangl2a mutants

Is the increase in transition states in Figure 2c due to the collapse of the heart or to the mutation itself?

Figure 2h

high variability in phenotypes; again, mutants should be used.

Figure 4a

single plane images at the mid-sagittal level should help to check the basolateral localization of N-cadherin; and higher magnification images should be provided.

Figure 4e

The text states that loss of *fzd7a* leads to apical constriction of cardiomyocytes at 54 hpf. Hard to see this claim from the data shown.

Fig 4d-f

quantification needed in terms of the number of disordered CMs and apical constrictions

Figure 4g-n

hard to understand this figure. The authors have just reported the observations, but there is no explanation of their meaning/significance. For e.g. why are certain transition-states devoid of pMRLC (line 222)? Why is there differential co-localization in ventricle vs. OFT (49% vs. 68%)? Further, they claim that they quantified the data on apical tight junctions (line 224); but there is no staining for tight junction markers.

Figure Supplementary 3

Why the ROCK inhibitor does not have a uniform effect in reducing pMRLC in all the CMs?

Figure 4o

Mutants should be used and images should be included showing the phenotype of all genotypes.

Figure 5

It is not clear that actin and pMRLC are planar polarised on a tissue scale. The Alcam staining (Fig 5 a-d) also appears enriched in the distal ventricle and the OFT compared to the proximal region and thus the enrichment doesn't seem to be specific for pMRLC/F-actin.

Figure 5e-h.

the stainings showing the switch in the localization of actin and myosin from basal in AV to apical in the OFT are not clear. Actin does not seem to be basal in the AV region in the picture they show and myosin seems to be apical and basal in the OFT region where they claim it should be apical

further reservations regarding the interpretation of results from Fig 5 e-h.

- First, the authors have not done any experiment to prove that the AV/proximal region of the ventricle has higher basal tension and distal/OFT region has higher apical tension.

- As I understand, their interpretation is based on the differential localization of pMRLC/ F-actin in the AV vs. OFT region, which itself is not very clear. By this stage, the looping has happened and depending on which part of the heart they are imaging and the angle of imaging, the localization pattern can be different. Further, the absence of this polarized actomyosin localization in *fzd7a* and *vangl2* deficient heart could just be because the shape of the organ is strikingly different in these conditions.

- Further, to differentiate apical vs. basal localization based on Alcam staining is not reasonable. The authors should take advantage of published cardiomyocyte specific polarity lines (Jimenez et al, 2016, Cell Reports) to differentiate apical vs. basal domain and provide some high-resolution images to help the readers understand this figure.

Figure 6e-g

Again the stainings are not very convincing or maybe it just that the picture is not representative. They claim that in *fzd7a* morphants the actomyosin localization is not polarized anymore but in image 6f and 6g one can observe some kind of polarization in the OC region.

Also, the alcam staining to label the CM membranes is not working properly when they use the *fzd7a* and *vangl2a* morphants (Figure 6e and 6q). So, if the membrane staining is not working properly it is difficult to believe the mislocalization of actin and myosin in these embryos.

Reviewer #3:

Remarks to the Author:

In this manuscript, the authors present a careful descriptive study of cardiac morphogenesis related to heart looping, towards a better understanding of the cellular basis for chamber ballooning as the primitive linear heart tube transitions. They use the zebrafish model starting at the 22 hpf linear heart tube and image individual CMs in the heart tube primarily through 54 hpf, to measure polarity and cell behaviors. Their first observation is that CMs form transition states in the epithelium, as is typical for many tube-based organ systems, as numerous cells initially have 4 or more neighbors and during this period exchange neighbors to form new defined boundaries, with about 1/10 cells in TS at 26 hpf and only 1/5 in TS at 54 hpf. Based on other systems, they investigate PCP as a likely regulator of this epithelial morphogenesis, and show that indeed in double mutants/morphants for Wnt5b and Wnt11, approximately 1/10 cells remain in TS at 54 hpf. The rest of the manuscript attempts to correlate this feature with downstream effectors. No changes in N-Cadherin localization are noted, but they find alterations in colocalization of actomyosin activity (F-actin-binding-GFP plus p-MRLC) that tends to colocalize with TS cells in wildtype, but less so in Fzd7a or Vangl2 knockdowns. A second observation is that actomyosin activity is also planar-polarized at the tissue level, with basal localization at the AV/proximal side, and apical localization in the OFT/distal region. This remains primarily basal throughout the heart tube in Fzd7a and Vang2 morphants and correlates with disruption of cardiac looping in these embryos. This is an excellent use for the zebrafish model, and a nice attempt at careful quantitative analysis of CM behavior during heart tube looping. There are a few issues that need attention and/or clarification.

- 1) A major weakness of the study is the inconsistent use of mutants/morphants for different aspects of the study. The initial observations suggested a convincing requirement for both Wnt5b and Wnt11 for resolution of TS cell behavior, but these mutants are never analyzed again in the rest of the manuscript, focusing instead mainly on Fzd7a and Vangl2 morphants. Do the Wnt5b/Wnt11 double mutants show the same loss of tissue polarity for actomyosin activity?
- 2) Loss of Fzd7a is reported to increase TS cells, but this is seen already at 26 hpf, not during the process of heart tube looping. Therefore, is this process occurring earlier than described, rather than during looping stages?
- 3) Early in the study there is no significant change in TS cell numbers for either the Vngl2 or Pk1a deficient embryos. These are thought to be negative regulators of the PCP program, so why is TS not enhanced compared to WT? And if this is the case, does this mean that TS cells are not relevant to cardiac looping, since the authors use the loss of Vngl2 to study the roll of PCP and looping in the rest of the study?
- 4) An example of selected images is shown in Fig. 4ef, where individual cells look markedly disturbed. However, just a single cell is shown. Are these fully representative, and how many cells were imaged?
- 5) Again, in terms of actomyosin activity, the Vangl2 knockdown shows a complete absence of TS in the OFT. Why is this, since the original argument would have predicted the opposite of a Fzd7a mutant phenotype.
- 6) With respect to planar-polarization at the tissue level, basal localization in the AV/proximal region is presented (not very convincingly) in Fig. 5. However, this is indicated as true for 4/8 heart tubes, with only a single trace, which is presumably the most obvious. Does this mean it is essentially random, since 4/8 did not show this pattern? Why is the pattern abolished in both the Fzd7a and Vangl2 morphants, rather than showing opposite effects?

7) At no time do the authors demonstrate that these treatments actually alter PCP signaling (especially important for morphants).

8) The alteration in heart tube looping is only shown for morphants in the last figure. These should be repeated using validated mutants, since morpholinos are notorious for causing non-specific heart looping defects.

Point-by-point response to the referees' comments

Reviewers' comments:

Reviewer #1 (Remarks to the Author):

This paper by the group of Daniela Panakova addresses the role of Planar Cell Polarity (PCP) signaling during the remodeling of the heart tube in zebrafish. The authors analyze the morphogenetic cellular re-arrangement that take place between the liner heart tube stage and the mature heart and how the Wnt5/11 and core Frizzled (Fz)-PCP factors influence the process. Overall the data are of high quality and are presented adequately. With some revisions (see below) the paper is worth considering for Nature Communications and will make an exciting contribution to the field and to PCP-regulated morphogenesis processes in general

There are a few issues with the paper, as currently presented, that should be addressed before the paper is published.

Specific comments:

i) While the data presentation overall is adequate, there are some instances where better data should be added. For example, all analyses of Vangl2 in figures 6 and 7 rely on morphant experiments, there are Vangl2 mutant alleles/null alleles (used in this study and otherwise) and these are better tools.

We thank the reviewer for the constructive feedback. In the revised version, we analyzed side-by-side *fzd7a* and *vangl2* morphants and mutants. We added new data in Figure 4, 5, 6, and 7.

ii) The presentation of the data figures could be arranged differently to allow direct comparison of wild-type vs the individual mutant scenarios. As it is now Figure 1 is all wild-type while figures 2-3 is all mutant. It would be better to align wild-type with mutant for direct comparison and split the figure panels into figures along the assays or measurement types used in analyses. Similarly, Figures 5-6 should be merged to allow direct comparison (with possibly some aspects going to a Supplemental Figure).

To allow direct comparison between wild type and mutant phenotypes we now merged or split the figures according to the assay types with the exception of the analysis of cellular orientation.

iii) In general terms, the text is very descriptive with many, many details stated in the main text, including many percentages of effects and associated statistical analyses, and then again repeated in the figure legends and also shown in the figures. This applies to the whole Results section, for example lines 128-133 or lines 141-148, or 1154-160, or 169-172, or 180-183. It is very difficult to read with all the numerical values inserted in the main text.

We fully agree with the reviewer and have omitted most of the numerical values from the main text to ease the readability, while providing the details especially regarding the statistical analysis in the Figure legend and/or Figure panels.

iv) In the Introduction on PCP, lines 65-72, the authors just list one review from Yingzi Yang as reference. While that review is fine, the authors should remember that all the PCP knowledge there is, it comes from work in *Drosophila* (without which the authors would not

be able to write the paper...). As such they should recognize the Drosophila work as basis for their own and cite a few recent reviews from the key Drosophila labs in the field. Examples to cite would be:

- Singh J, and Mlodzik M. (2012). Planar cell polarity signaling: coordination of cellular orientation across tissues. Wiley Interdiscip Rev Dev Biol. 2012 Jul-Aug;1(4):479-99.
- Goodrich LV, and Strutt D. (2011). Principles of planar polarity in animal development. Development. 2011 May;138(10):1877-92.
- Adler PN. (2012). The frizzled/stan pathway and planar cell polarity in the Drosophila wing. Curr Top Dev Biol. 2012;101:1-31.
- Fanto M, McNeill H. (2004). Planar polarity from flies to vertebrates. J Cell Sci. 2004 Feb 1;117:527-33.
- Bayly R, Axelrod JD. (2011). Pointing in the right direction: new developments in the field of planar cell polarity. Nat Rev Genet. 2011 Jun;12(6):385-91.

We apologize for inadvertently omitting citing the appropriate literature sources. We have now added the references to the revised manuscript.

v) Pge 9 Lines 212-214: The original work showing that Rho-family GTPases are involved in PCP signaling should be included:

- Eaton S, Wepf R, and Simons K. (1996). Roles for Rac1 and Cdc42 in planar polarization and hair outgrowth in the wing of Drosophila. J Cell Biol. 1996 Dec;135(5):1277-89.
- Strutt DI, Weber U, and Mlodzik M. (1997). The role of RhoA in tissue polarity and Frizzled signalling. Nature. 1997 May 15;387(6630):292-5.
- Boutros M, Paricio N, Strutt DI, and Mlodzik M. (1998). Dishevelled activates JNK and discriminates between JNK pathways in planar polarity and wingless signaling. Cell. 1998 Jul 10;94(1):109-18.

We apologize for not citing these original articles; we include now the relevant references in the revised manuscript.

--

Reviewer #2 (Remarks to the Author):

This paper deals with the interesting process of cardiac chamber ballooning, and more specifically with the behavior of the cardiomyocytes during this process. The authors put forth a model by which the Planar Cell Polarity (PCP) pathway coordinates the localization of actomyosin activity and thus the efficiency of cell neighbor exchanges. Further claims include tissue tension in the control of cardiac chamber ballooning as well as cardiac looping. While the questions addressed are important, there are a number of important issues with the current manuscript that need to be addressed in full before it can be considered further.

General comments:

1. The cell-transition experiments were done in non-contracting hearts; not clear how physiologically relevant these data are considering that these rearrangements will be strikingly different in the beating heart (as acknowledged in the MS). In other words, the experiments will need to be done again in contracting hearts.

We thank the reviewer for the critical comment. We have considered to image the whole process of cardiac chamber ballooning and looping *in toto*, and performed pilot experiments

together in collaboration with Prof. Jan Huisken, the prominent expert in the high-speed SPIM imaging. We have however not been successful obtaining the images with high enough spatial subcellular resolution. In our analysis, the transition state is defined as 3-pixel point in 512x512 pixel image; the current technology available to us does not unfortunately provide this resolution. While we agree that the dynamics of the cell neighbour exchange might be effected in *tnnt2* MO injected embryos as we point out in the text ourselves, the data shows that this process does occur during cardiac chamber remodelling. The aim of our work was not to focus on the dynamics of the process, but rather describing its presence in the myocardium, and related mechanism of regulation. We would like to argue that our detailed analysis at different stages of cardiac chamber formation provides sufficient evidence to conclude that cell neighbour exchange could contribute to cardiac chamber formation. We are currently developing strategies akin to optogenetic control to decouple excitation-contraction coupling to undertake the whole analysis in living embryos, which is a focus of another major study, and ongoing research in the lab.

2. The authors equate Wnt5b and Wnt11 with PCP signaling, a now discounted simplification (<http://web.stanford.edu/group/nusselab/cgi-bin/wnt/receptors>)

We apologize for this unintended simplification, and edited the sentence:

“Wnt5b and Wnt11 belong to key Wnt ligands required in cardiogenesis that have been described in the context of the PCP signalling pathway.”

3. morphants and mutants: some of the phenotypes are fairly subtle (and variable), thus one should take advantage of the blinding aspect of a genetic approach to double check claims (ie, cross two hets together, analyze the embryos, sort them according to phenotype and then genotype).

We thank the reviewer for the comment. We have now performed all the experiments both in mutants and morphants, resulting in new Figure 4, 5, 6, and 7 (the cell neighbor exchange assays have been performed in both mutant and morphants already in the original submission). All morphant data are now presented in the supplement for the comparison. In addition, we performed in depth analysis of the mutant phenotype penetrance, including the blinded genotyping experiments. As per current guidelines (Stainier DYR *et al*, 2017), we have shown that morpholinos upon injections into corresponding mutants do not result in additional defects. These defined MO concentrations were used in all experiments. All these data are summarized in Supplementary Figure 1.

4. The manuscript is not easy to read, some images are of low quality and/or insufficient magnification, some key references appear to be missing, and the work would benefit from using some recently published reagents (including transgenic lines).

We have edited the manuscript text to ease the readability; the numerical values are now presented in the Figure legend or directly in the Figure panels. We replaced all the images of low quality with high-resolution data. We added references, especially the ones regarding the original work describing involvement of Rho-GTPases in PCP signaling. We in-crossed *fzd7a* and *vangl2* mutants with *Tg(myl7:LifeAct-GFP)* to facilitate the analysis of actin localization, see also below in other comments.

Other comments:

Figure 1a, 1b and 1c

Please show the different cell shapes at IC and OC and cell orientation.

We now provide the updated schematic in Figure 1a with labeled IC, OC, AV, and OFT cells and their orientation within the ventricle.

Based on the drawing, the orientation (Fig 1b) of the CMs in the AV canal is not clear; transition states and 3-point vertices were not precisely marked in all conditions.

We used published Packing analyser software v2.0 followed by the custom MatLab based software (which algorithm is described in detail in the method section) to analyse the transition states and the cell orientation, respectively. Using the software we have minimized human-based errors; the transition states and 3-point vertices were marked by the software as opposed to manual labeling. The orientation of AV cells is now labeled in Figure 1a.

check effect on transition states in *wnt5b*, *wnt11*, *fzd7a* and *vangl2a* mutants

The transition states in *wnt5b*^{ta98} *-/-*, *wnt11*^{tx226} *-/-*, *wnt5b*^{ta98}; *wnt11*^{tx226} *-/-*, *fzd7a*^{e3} *-/-* and *vangl2*^{m209} *-/-* mutants were analysed and presented already in the original submission. We now label the mutant alleles more clearly, and in the graph of quantification of transition states in Figure 21 (originally Figure 2h) we use color code to distinguish better between mutant and morphant phenotypes.

Is the increase in transition states in Figure 2c due to the collapse of the heart or to the mutation itself?

Indeed, the double *wnt5b*; *wnt11* morphant as well as mutant embryos are smaller as is the size of their hearts. The number of transitory states is however quantified over 100 ventricular cardiomyocytes to address exactly this issue, which allows for comparison of hearts of different sizes.

Figure 2h
high variability in phenotypes; again, mutants should be used.

Figure 2h in the originally submitted manuscript is now Figure 21. We used following published mutants: *wnt5b*^{ta98} *-/-*, *wnt11*^{tx226} *-/-*, *wnt5b*^{ta98}; *wnt11*^{tx226} *-/-*, *fzd7a*^{e3} *-/-*, and *vangl2*^{m209} *-/-*. The graph is plotted together with data obtained from corresponding morphant phenotypes. For clarity, the color code is used to distinguish between mutants and morphants.

Figure 4a
single plane images at the mid-sagittal level should help to check the basolateral localization of N-cadherin; and higher magnification images should be provided.

We now provide the data displaying N-cadherin localization in hearts of wild type embryos (n=19), and compare them to *fzd7a*^{e3} (n=15) and *vangl2*^{m209} (n=14) mutant hearts in top-down as well as mid-sagittal sections. The mid-sagittal sections are co-stained against aPKC, an apical tight junction marker to visualize the localization of N-Cadherin along whole of lateral membranes as oppose to distinct apical localization of aPKC. All images are provided in higher resolution (format: 1024x1024, 8bit). The data showing N-Cadherin localization in hearts of *fzd7a* and *vangl2* MO-injected embryos are now presented in supplement.

Figure 4e

The text states that loss of *fzd7a* leads to apical constriction of cardiomyocytes at 54 hpf. Hard to see this claim from the data shown.

We now provide data from wild type, *fzd7a^{e3}*, and *vangl2^{m209}* mutant hearts expressing transiently membrane-bound GFP. In addition to depth projections, we now also provide mid-sagittal sections to improve the visualization. As not all of *fzd7a^{e3}* mutant cells appear apically constricted we corrected that in the text.

Fig 4d-f

quantification needed in terms of the number of disordered CMs and apical constrictions

We would like to report here just qualitative observations, and refrain from any quantification of cellular shapes upon loss of *fzd7a* and *vangl2*; due to high variability of GFP expression upon transient expression, the quantification (e.g. volumetric analysis) is not currently feasible.

Figure 4g-n

hard to understand this figure. The authors have just reported the observations, but there is no explanation of their meaning/significance. For e.g. why are certain transition-states devoid of pMRLC (line 222)? Why is there differential co-localization in ventricle vs. OFT (49% vs. 68%)? Further, they claim that they quantified the data on apical tight junctions (line 224); but there is no staining for tight junction markers.

We agree with the reviewer that it is intriguing that pMRLC is not always present at the transition states, and that it is localized differentially in the ventricular and OFT cardiomyocytes. We now provide the quantification obtained by analyzing *fzd7a* and *vangl2* mutants carrying *Tg(myl7:LifeAct-GFP)* that corroborate the observations of morpholino phenotypes in the original Figure 4o, and provide statistical data showing significance. The quantification of morpholino-based data is now presented in the supplement. We surmise that this is due to the dynamics of phosphorylation state of MRLC. We interpret it as PCP signaling regulating the dynamic of phosphorylation state of MRLC, based also on already published work that we now cite more precisely, see also in the response to the reviewer 1. Due to the number of proteins tested and available channels at the confocal microscope, we cannot unfortunately always use apical tight junction markers. Given however our experience with the model, we always present comparable sections inferred from such stainings as shown in Figure 4a.

Figure Supplementary 3

Why the ROCK inhibitor does not have a uniform effect in reducing pMRLC in all the CMs?

We now show the whole ventricle and not just ROI stained against pMRLC in DMSO- and Y-27623-treated hearts in supplement. We analyse mean fluorescence intensity of pMRLC in several hearts from 3 independent experiments showing the reduced signal upon ROCK inhibitor treatment.

Figure 4o

Mutants should be used and images should be included showing the phenotype of all genotypes.

We now provide the quantification obtained by analyzing *fzd7a* and *vangl2* mutants carrying *Tg(myl7:LifeAct-GFP)* that corroborate the observations of morpholino phenotypes in the original Figure 4o. Qualitatively, we have observed only two states in control as well as in

fzd7a and *vangl2* mutants, either pMRLC is present at the transition states or it is not. The difference is in the percentage of the colocalization with actin between distinct ventricular regions that we tested for statistical significance.

Figure 5

It is not clear that actin and pMRLC are planar polarised on a tissue scale. The Alcam staining (Fig 5 a-d) also appears enriched in the distal ventricle and the OFT compared to the proximal region and thus the enrichment doesn't seem to be specific for pMRLC/F-actin.

We now provide clearer labeling of proximal and distal ventricular regions in addition to atrioventricular junction that is delineated by differential staining of Alcam in ventricular vs atrial cardiomyocytes in addition to the morphological constriction that starts to be visible around 30 hpf in control hearts. In all 16 hearts imaged from 4 independent experiments we have only occasionally observed regional differences in Alcam localization in the control hearts.

Figure 5e-h.

the stainings showing the switch in the localization of actin and myosin from basal in AV to apical in the OFT are not clear. Actin does not seem to be basal in the AV region in the picture they show and myosin seems to be apical and basal in the OFT region where they claim it should be apical

We thank the reviewer for the feedback. We now performed the experiments in the mutant conditions: *wnt5b^{ta98}*; *wnt11^{tx226}*, *fzd7a^{e3}*, *vangl2^{m209}*, and increased the number of control hearts tested. Rather than switch in localization, we now describe the changes in localization of actin and phospho-MRLC as apical accumulation and randomized localization that more precisely describe our findings.

further reservations regarding the interpretation of results from Fig 5 e-h.

- First, the authors have not done any experiment to prove that the AV/proximal region of the ventricle has higher basal tension and distal/OFT region has higher apical tension.

We fully agree with the reviewer that we did not measure the tension in different regions of the ventricle. We can only infer that higher abundance of phospho-MRLC may result in changes in tissue tension. We corrected the interpretation in the manuscript text.

- As I understand, their interpretation is based on the differential localization of pMRLC/ F-actin in the AV vs. OFT region, which itself is not very clear. By this stage, the looping has happened and depending on which part of the heart they are imaging and the angle of imaging, the localization pattern can be different. Further, the absence of this polarized actomyosin localization in *fzd7a* and *vangl2* deficient heart could just be because the shape of the organ is strikingly different in these conditions.

We would like to argue that at 30 hpf the process of looping is initiated and only finished by 2 dpf, as seen in Figure 7a or reviewed in reference 2 (Bakkers J, 2011). As mentioned earlier however, in the revised manuscript we refrain from using the term "switch in localization". Our analysis of *wnt5b^{ta98}*; *wnt11^{tx226}*, *fzd7a^{e3}*, *vangl2^{m209}* mutants showed randomized localization of both actin and myosin rather than their apical accumulation in the distal ventricle and OFT region, and data obtained from corresponding morphants corroborate this interpretation. We agree that the shapes of the heart are different in all mutant conditions, but

we would like to argue that this is not due to technical inconsistencies. All samples for given biological replicate are handled on the same day, with the same conditions, the same laser settings are used for all samples that are imaged on the same day, and finally Z-projections are assembled from the corresponding sections, comparable between control and experimental samples. So rather than due to looping defects, we would like to argue that the randomized distribution of the actomyosin activity in the absence of PCP signaling represents the biological situation.

- Further, to differentiate apical vs. basal localization based on Alcam staining is not reasonable. The authors should take advantage of published cardiomyocyte specific polarity lines (Jimenez et al, 2016, Cell Reports) to differentiate apical vs. basal domain and provide some high-resolution images to help the readers understand this figure.

We now provide staining against apical tight junction marker aPKC in Figure 4a to clarify the topology of the myocardium. Unfortunately, due to number of protein tested and available channels at the confocal microscope, we cannot always use apical tight junction markers. The transgene, Podocalyxin:GFP, generated and described in Jimenez-Amilburu *et al*, 2016, labels apical membranes, but it is absent from apical tight junctions, thus it is not feasible to use in context of our experimental setup.

Figure 6e-g

Again the stainings are not very convincing or maybe it just that the picture is not representative. They claim that in *fzd7a* morphants the actomyosin localization is not polarized anymore but in image 6f and 6g one can observe some kind of polarization in the OC region.

We have performed the experiments in *fzd7a* and *vangl2* mutants, see above, and corroborated our results obtained from morpholino-treated embryos presented in the original submission showing what we now refer to as randomized localization.

Also, the alcam staining to label the CM membranes is not working properly when they use the *fzd7a* and *vangl2a* morphants (Figure 6e and 6q). So, if the membrane staining is not working properly it is difficult to believe the mislocalization of actin and myosin in these embryos.

We have revised these experiments, and hope we now provide more satisfactory data.

--

Reviewer #3 (Remarks to the Author):

In this manuscript, the authors present a careful descriptive study of cardiac morphogenesis related to heart looping, towards a better understanding of the cellular basis for chamber ballooning as the primitive linear heart tube transitions. They use the zebrafish model starting at the 22 hpf linear heart tube and image individual CMs in the heart tube primarily through 54 hpf, to measure polarity and cell behaviors. Their first observation is that CMs form transition states in the epithelium, as is typical for many tube-based organ systems, as numerous cells initially have 4 or more neighbors and during this period exchange neighbors to form new defined boundaries, with about 1/10 cells in TS at 26 hpf and only 1/5 in TS at 54 hpf. Based on other systems, they investigate PCP as a likely regulator of this epithelial morphogenesis, and show that indeed in double mutants/morphants for *Wnt5b* and *Wnt11*, approximately 1/10 cells remain in TS at 54 hpf. The rest of the manuscript

attempts to correlate this feature with downstream effectors. No changes in N-Cadherin localization are noted, but they find alterations in colocalization of actomyosin activity (F-actin-binding-GFP plus p-MRLC) that tends to colocalize with TS cells in wildtype, but less so in *Fzd7a* or *Vangl2* knockdowns. A second observation is that actomyosin activity is also planar-polarized at the tissue level, with basal localization at the AV/proximal side, and apical localization in the OFT/distal region. This remains primarily basal throughout the heart tube in *Fzd7a* and *Vangl2* morphants and correlates with disruption of cardiac looping in these embryos. This is an excellent use for the zebrafish model, and a nice attempt at careful quantitative analysis of CM behavior during heart tube looping. There are a few issues that need attention and/or clarification.

1) A major weakness of the study is the inconsistent use of mutants/morphants for different aspects of the study. The initial observations suggested a convincing requirement for both *Wnt5b* and *Wnt11* for resolution of TS cell behavior, but these mutants are never analyzed again in the rest of the manuscript, focusing instead mainly on *Fzd7a* and *Vangl2* morphants. Do the *Wnt5b/Wnt11* double mutants show the same loss of tissue polarity for actomyosin activity?

We thank the reviewer for the constructive comments. As mentioned in the response to reviewer 1 and 2, we have now probed both major observations: the effect on cell neighbor exchange as well as on the polarization of the actomyosin activity both in *wnt5b^{ta98}*; *wnt11^{tx226}* double mutants as well as *fzd7a^{e3}*, and *vangl2^{m209}* mutants; for actomyosin assays we generated *fzd7a^{e3}*, and *vangl2^{m209}* mutant lines carrying *Tg(myl7:LifeAct-GFP)*. These data corroborate our observations from the corresponding morphants originally submitted. As a result, Figure 4, 5, 6, and 7 include new data; all morphant data are now presented in the supplement for the comparison. In addition, we performed in depth analyses of the mutant phenotype penetrance, including the blinded genotyping experiments. As per current guidelines (Stainier DYR *et al*, 2017), we have shown that morpholinos upon injections into corresponding mutants do not result in additional defects. These defined MO concentrations were used in all experiments. All these data are summarized in Supplementary Figure 1.

Interestingly, *wnt5b^{ta98}*; *wnt11^{tx226}* double mutants do not seem to effect the actomyosin activity locally at the transition states, but they do show similar defect in the tissue-scale polarization of actomyosin activity as *fzd7a^{e3}*, and *vangl2^{m209}* mutants. As to why, we plan to address this in the future studies.

2) Loss of *Fzd7a* is reported to increase TS cells, but this is seen already at 26 hpf, not during the process of heart tube looping. Therefore, is this process occurring earlier than described, rather than during looping stages?

This is a very important point, for which we unfortunately do not have sufficient data at this moment to elaborate. It is very well possible that *Fzd7a* plays a role already during the migration of the bilateral heart fields and/or during the assembly of the cardiac disc. These questions go, however, beyond the aim of this study, which was to address the processes regulating heart tube remodeling. We are developing strategies to address these important aspects of cardiac biology in the future studies.

3) Early in the study there is no significant change in TS cell numbers for either the *Vngl2* or *Pk1a* deficient embryos. These are thought to be negative regulators of the PCP program, so why is TS not enhanced compared to WT? And if this is the case, does this mean that TS cells are not relevant to cardiac looping, since the authors use the loss of *Vngl2* to study the roll of PCP and looping in the rest of the study?

We thank the reviewer for bringing this observation up. As in the previous response, we do not have currently enough data to explain the mild effect of Vangl2/Pk1 loss. It is possible that due to the dynamic state of TS, the net number of TS seem to be unchanged. As argued above, to capture the dynamic of cell neighbor exchange, we are developing strategies to visualize the process in living embryos. To obtain high-resolution data, we are working on the methodologies to decouple excitation-contraction coupling akin to optogenetic tools. Nevertheless, we would like to argue that our data add to the current two-step model of the cardiac chamber formation and looping. We argue that cell neighbor exchange is required to expand the chambers, most likely facilitated by the local actomyosin activity, while the looping process requires tissue-scale polarization of actomyosin; in both instances PCP contributes to the process. To reconcile our use of morpholinos, we repeated the experiments addressing the actomyosin polarization in *vangl2* mutants (and *fzd7a* mutants, and *wnt5b;wnt11* double mutants) as mentioned above.

4) An example of selected images is shown in Fig. 4ef, where individual cells look markedly disturbed. However, just a single cell is shown. Are these fully representative, and how many cells were imaged?

We now present both, single cells as depth projections and group of cells in mid-sagittal sections from ventricular outer curvatures of hearts of *fzd7a* and *vangl2* mutants, in which we transiently express *myl7:lck-EGFP*. We wish to report only on the qualitative aspects of loss of *fzd7a* and *vangl2* that lead to formation of basal protrusions, and disturbed basal membrane, respectively. All together we imaged 19 wt hearts, 12 *fzd7a* mutant hearts and 11 *vangl2* mutant hearts from at least two biological replicates with several individual cells or cluster of cells.

5) Again, in terms of actomyosin activity, the Vangl2 knockdown shows a complete absence of TS in the OFT. Why is this, since the original argument would have predicted the opposite of a *Fzd7a* mutant phenotype.

We see the complete absence of TS in *vangl2* mutant hearts at 30 hpf, but not at 54 hpf at the end of the process of cardiac chamber expansion and looping. Again, the live imaging would be very useful to be able to explain these finding, and we are working on the methodologies to do it. We hope our future studies will reconcile these observations. In addition, even though Vangl2/Pk1 antagonizes Fzd7/Dvl, the relationship is not always so linear. As much as we would like to understand these relationships, currently this goes beyond the scope of this study.

6) With respect to planar-polarization at the tissue level, basal localization in the AV/proximal region is presented (not very convincingly) in Fig. 5. However, this is indicated as true for 4/8 heart tubes, with only a single trace, which is presumably the most obvious. Does this mean it is essentially random, since 4/8 did not show this pattern? Why is the pattern abolished in both the *Fzd7a* and *Vangl2* morphants, rather than showing opposite effects?

We thank reviewer for this comment. We have now analyzed 16 control hearts in 4 independent experiments. Indeed, we observe that in the inner curvature of the OFT the apical accumulation of actin is randomized, this is somewhat true for pMRLC as well. However, there is a clear bias towards apical accumulation of both actin and pMRLC in the outer curvature of the OFT. We refrain from using the term “switch” in favor of apical accumulation vs randomized localisation in the revised version of the manuscript. In addition, we labeled more clearly the proximal and distal ventricular regions in the figure panels.

Our current data cannot fully explain why this process is randomized in *fzd7a*, *vangl2*, *wnt5b*; *wnt11* mutants. One possibility that we would like to explore further is the differential effect on myosin phosphorylation state regulated by Fzd/Dvl branch via e.g. ROCK and actin polymerization regulated by Vangl2/Pk1 through formins. Our future studies will attempt to address these relationships further.

7) At no time do the authors demonstrate that these treatments actually alter PCP signaling (especially important for morphants).

While we do not study directly the effectors of PCP signaling, we do show the effects on actomyosin activity that lays downstream of these effectors in the signaling cascade. We perform all the experiments in the mutants in the revised manuscript; we show that morpholino injection does not result in additional phenotypes in the *wnt5b*, *wnt11*, *fzd7a*, and *vangl2* mutants suggesting the specificity of the morpholinos as per current guidelines (Stainier DYR *et al*, 2017).

8) The alteration in heart tube looping is only shown for morphants in the last figure. These should be repeated using validated mutants, since morpholinos are notorious for causing non-specific heart looping defects.

We thank the reviewer for raising this point. We have now measured the looping angle in both *fzd7a* and *vangl2* mutants carrying *Tg(myf7:LifeAct-GFP)* transgene. In addition we also performed the *in vitro* looping assay in these mutants.

Reviewers' Comments:

Reviewer #1:

Remarks to the Author:

The revised paper by Merks et al (Panakova lab) is significantly improved. The authors have addressed my comments and, as far as I can tell, the comments from the other reviewers satisfactorily. I recommend publication of the revised paper in Nature Communications.

Reviewer #2:

Remarks to the Author:

The extensive revisions have certainly made the paper stronger (ie, technically more sound).

The schematic in Figure 6c is very helpful and additional schematics should help make the paper more approachable (especially when the point the authors are trying to make is not so obvious from the data shown/resolution used (eg, some of the immunostainings))

It is unfortunate that the authors were not able to image the various cellular processes of interest in contractile hearts as they are most likely to happen differently in non-contractile hearts; additional emphasis should be given to this important point (including additional references).

Reviewer #3:

Remarks to the Author:

The revised manuscript by Merks et al. has been improved, primarily by validation of results using mutants in addition to the morphants, and by rearrangement of some figure panels. Overall it is a rigorous and interesting study of PCP component regulation of cardiac morphogenesis.

However, for several of my key points, the authors "hand-waved". I think it is justifiable to suggest that solving these issues is "beyond the scope" of the current study, although it does not seem reasonable to just ignore them. I can imagine readers will have the same queries. I suggest that the authors at least point out the limitations of their analysis, for example in the discussion. Two responses in particular that could be addressed in the discussion are:

"This is a very important point, for which we unfortunately do not have sufficient data at this moment to elaborate. It is very well possible that Fzd7a plays a role already during the migration of the bilateral heart fields and/or during the assembly of the cardiac disc."

"As in the previous response, we do not have currently enough data to explain the mild effect of Vangl2/Pk1 loss. It is possible that due to the dynamic state of TS, the net number of TS seem to be unchanged."

Point-by-point response to the reviewers' comments - 2nd revision

REVIEWERS' COMMENTS:

Reviewer #1 (Remarks to the Author):

The revised paper by Merks et al (Panakova lab) is significantly improved. The authors have addressed my comments and, as far as I can tell, the comments from the other reviewers satisfactorily. I recommend publication of the revised paper in Nature Communications.

We thank the reviewer for the positive feedback.

--

Reviewer #2 (Remarks to the Author):

The extensive revisions have certainly made the paper stronger (ie, technically more sound).

We thank the reviewer for the positive feedback.

The schematic in Figure 6c is very helpful and additional schematics should help make the paper more approachable (especially when the point the authors are trying to make is not so obvious from the data shown/resolution used (eg, some of the immunostainings))

We provide the schematics in Figure 1a and Figure 6c only, but where appropriate we added arrowheads or asterisks to guide the reader. We are not sure at this point, which immunostainings are not easy to follow.

It is unfortunate that the authors were not able to image the various cellular processes of interest in contractile hearts as they are most likely to happen differently in non-contractile hearts; additional emphasis should be given to this important point (including additional references).

As in our previous response, we agree with the reviewer's concerns. We would like to emphasize that in non-contratile hearts, we observe the resolution of the transition states similarly as reportedly observed in other epithelia, as we could observe the cell boundaries shrinking and expanding. We would like to argue that most likely effect of non-contractile heart is on the timing; while in other epithelia the transition state resolution happens within minute range, we observe the process over several hours. We now discuss this point in the discussion more extensively as a caveat of our findings.

--

Reviewer #3 (Remarks to the Author):

The revised manuscript by Merks et al. has been improved, primarily by validation of results using mutants in addition to the morphants, and by rearrangement of some figure panels. Overall it is a rigorous and interesting study of PCP component regulation of cardiac morphogenesis.

We thank the reviewer for the positive feedback.

However, for several of my key points, the authors "hand-waved". I think it is justifiable to suggest that solving these issues is "beyond the scope" of the current study, although it does not seem reasonable to just ignore them. I can imagine readers will have the same queries. I suggest that the authors at least point out the limitations of their analysis, for example in the discussion. Two responses in particular that could be addressed in the discussion are:

"This is a very important point, for which we unfortunately do not have sufficient data at this moment to elaborate. It is very well possible that Fzd7a plays a role already during the migration of the bilateral heart fields and/or during the assembly of the cardiac disc."

"As in the previous response, we do not have currently enough data to explain the mild effect of Vangl2/Pk1 loss. It is possible that due to the dynamic state of TS, the net number of TS seem to be unchanged."

We apologize if our arguments sounded "hand-waving". We fully agree with the raised points, and discuss the shortcomings of our approach with what we now believe is more appropriate response.

The response to the both reviewers #2 and #3 is added as a full new paragraph in the Discussion:

Cell neighbour exchange is a dynamic process that occurs within a range of minutes²⁷. Live imaging of non-contractile hearts (Supplementary movie 1) revealed that the cell neighbour exchange indeed occurs during chamber formation, with the caveat that the timing of cell boundaries shrinking and expanding is skewed. While the cardiac chambers form and the LHT remodels, the heart already beats at around 100 times per minute, hindering the high-resolution imaging required to attain high spatial resolution data at the subcellular level. The recent advancements in the field are on track to soon address this issue³⁰. As we observed the effect of PCP signalling on epithelial heart remodelling at steady state, we focused not on the dynamics of the process, but rather on its hallmark represented by the presence of the TS in the tissue. While we are unable to definitively conclude how PCP signalling regulates the resolution of the transitions states, we clearly observe that the loss of Wnt/Fzd7 signalling axis leads to marked accumulation of the TS in the tissue. The increased number of TS already at LHT in the *fzd7*-deficient hearts further indicates that Fzd7 may play a role already during the migration of the bilateral heart fields and/or during the LHT assembly. Nevertheless, the reduction in TS between 26 and 54 hpf in the *fzd7*-deficient hearts, albeit modest, suggests that Fzd7 is required also for their resolution during chamber expansion, and not only prior to the LHT formation. In contrast, the loss of Vangl2/Pk1 signalling axis has very mild effect on the accumulation of the TS in the tissue. Whether this is due to the highly dynamic nature of TS yielding unaltered net number of TS needs to be further determined. The ratio between the number of TS at 26 hpf to 54 hpf is 1.6, 1.4, and 2.1 in wild type, *fzd7a*-, and *vangl2*-deficient hearts, respectively, suggesting that the TS resolution is slower in the absence of *fzd7a* and faster in the loss of *vangl2*, and warrants further examination.